

# Metallic elements and oxides and their relevance to Laurentian Great Lakes geochemistry

Malachi N. Granmo, Euan D. Reavie, Sara P. Post and
Lawrence M. Zanko

Natural Resources Research Institute, University of Minnesota Duluth, Duluth, MN, USA

## ABSTRACT

The Laurentian Great Lakes are the most studied system in lake geochemistry and have well-preserved chronological profiles. Metals play numerous critical roles in natural and anthropogenic characteristics of lake ecosystems, so patterns in the historical records of metals from sedimentary cores provide important information about environmental baselines and human impacts. Relevant studies of Great Lakes geochemistry are listed, and we follow with encyclopedic descriptions of metals and their oxides in the lakes. These descriptions include likely natural and anthropogenic sources of elements, their known history from previous paleoecological studies, and their status as potential contaminants of concern. Despite the well-studied geology of the Great Lakes catchment, sourcing elements was sometimes difficult due to materials often being moved long distances by glaciation and the global prevalence of atmospheric pollutants. We summarized available information on metals and their roles as geochemical indicators in the Great Lakes.

## INTRODUCTION

An understanding of past conditions is essential for potential remediation efforts in an aquatic system because it can be used to estimate natural baseline, remediation targets, timing and causes of impacts, and positive effects of existing remedial efforts. During the Anthropocene epoch, human impacts have far exceeded natural changes in the landscape through agriculture and industry, mobilizing sediments and releasing pollutants into the air and water. These activities leave geochemical markers in sedimentary records of aquatic systems. Sediment cores can be collected and dated to reveal local geochemical histories. Metals are of particular interest because they record a history of pollution going back thousands of years (*Heim & Schwarzbauer, 2012*). In order to determine the historical extent of contamination caused by human activity, concentrations of potential pollutants must be compared to the baseline occurrences from natural sources (*Alderton, 1985*). Geochemistry depends on location-specific soil and bedrock composition, so it is important to consider natural sources of inorganic materials relative

Corresponding author
Malachi N. Granmo, aliff002@d.umn.edu

to anthropogenic sources. As *Tourtelot (1971)* describes, "[t]he types of rocks that form geologic units in the Earth's crust supply most of the raw materials from which soils are formed and from which water derives its inorganic constituents."

Sediments in lake systems are especially useful in geochemical studies because lakes act as sinks for inorganic materials. The best environments for geochemical studies are aquatic environments that have stable sedimentary basins allowing steady deposition of fine-grained material suitable for fixation of pollutants. Ideally, sediments are reducing (anoxic) at the sediment/water interface and thereby support minimal post-depositional mobility. According to *Heim & Schwarzbauer (2012),* the Laurentian Great Lakes are the most studied system in lake geochemistry and they have well-preserved chronological profiles (*Reavie et al., 2017*). These studies reveal a history of contamination starting with European settlement around 1850 and increasing over the next century with increases in industry, forest fires, and the burning of fossil fuels, which supported increased flux of land-based and atmospheric pollutants to the lakes. More recently, environmental regulations such as the Clean Water Act (*United States of America, 1972*) and the Great Lakes Water Quality Agreement (*Canada & United States of America, 1972*) have mitigated inputs of these pollutants and have also led to their remediation to some degree. A detailed list of pertinent studies along with associated analytes is provided in Table 1.

## Survey methodology

This review summarizes what is known of inorganic paleogeochemistry in the Laurentian Great Lakes, itemized by metallic elements. To date, the Great Lakes system has been considered in parts (Table 1), so we aimed to synthesize all relevant literature to create a central understanding of metallic geochemistry in the lakes. This review is the first of a pair of manuscripts exploring this topic, the second being a retrospective geochemical analysis of 11 sediment cores collected from the lakes (*Aliff et al., 2020*). Here we provide an overview of the metallic elements and oxides with a focus on their relevance in the Laurentian Great Lakes. Major aims were to define the analytes in terms of their sources and characterize their potential importance as anthropogenic stressors.

## Geology and geomorphology of the Great Lakes Basin

The geology of the Great Lakes basin has been extensively studied. Broadly, the basin is bounded to the north by the upland Precambrian-age Canadian Shield, which is dominated by intrusive rocks (Fig. 1).

The basin is bound to the south by a lowland region of Paleozoic sedimentary rocks such as limestones, dolomites, and sandstones (Fig. 1; *Larson & Schaetzl, 2001*). The basins of lakes Erie and Michigan, as well as most of the Lake Ontario basin, are in this lowland region. The Lake Superior basin is almost entirely in the Canadian Shield, and Lake Huron straddles the two.

The Canadian Shield is the exposed portion of the North American craton, which forms the core of the continent. "The Canadian Shield region north of Lake Superior features a series of Greenstone belts in Archean rocks up to 2.5 billion years. … Near the bottom of typical greenstone strata are ultrabasic volcanic and intrusive (rocks), which can yield

**Table 1** All known geochemical studies in the Laurentian Great Lakes, including the lengths of sedimentary profiles and metallic elements and oxides considered.

| References | Location | Max core length (time period) | Analytes |
|---|---|---|---|
| *Nussmann (1965)* | Lake Superior | 20 cm (1962) | Al, As, Ba, Be, Cd, Co, Cr, Cu, Fe, Li, Mn, Mo, Ni, Pb, Sb, Sn, Sr, V, Zn |
| *Callender (1969)* | Lake Michigan and Superior | 80 cm (1968) | Ca, Fe, Mg, Mn |
| *Ruch, Kennedy & Shimp (1970)* | Southern Lake Michigan | 325 cm (1969) | As |
| *Schleicher & Kuhn (1970)* | Southern Lake Michigan | 325 cm (1969) | P |
| *Shimp, Leland & White (1970)* | Southern Lake Michigan | 325 cm (1969) | Al, Be, Ca, Cr, Fe, K, Mg, Mn, Ni, P, Pb, Si, Ti, V |
| *Kennedy, Ruch & Shimp (1971)* | Southern Lake Michigan | 325 cm (1969) | Hg |
| *Shimp et al. (1971)* | Southern Lake Michigan | 325 cm (1969) | B, Be, Br, Cr, Co, Cu, La, MnO, Ni, Pb, Sc, V, Zn |
| *Cronan & Thomas (1972)* | Lake Ontario | 100 cm (1970) | Al, Ca, Fe, K, Mn, P, S, Ti |
| *Kemp, Gray & Mudrochova (1972)* | Lakes Ontario, Erie, and Huron | 50 cm (1800–1970) | C, N, P, S |
| *Kovacik (1972)* | Western Lake Erie | 60 cm (1971) | Hg |
| *Mothersill & Fung (1972)* | Northern Lake Superior basin | 738 cm (11,270 yr BP – 1971) | Ca, Cr, Cu, Fe, Mn, Ni, Sr, Zn |
| *Thomas (1972)* | Lake Ontario | 50 cm (1968) | Hg |
| *Walters et al. (1972)* | Western Lake Erie | 117 cm (1971) | Hg |
| *Williams & Mayer (1972)* | Lakes Erie and Ontario | 15 m (~10,000 yr BP–1971) | P |
| *Leland, Shukla & Shimp (1973)* | Southern Lake Michigan | 100 cm (1969) | Br, Cr, Cu, Pb, Zn |
| *Edgington, Robbins & Karttunen (1974)* | Lake Michigan | 20 cm (1830–1972) | Pb |
| *Kemp et al. (1974)* | Lakes Ontario, Erie, and Huron | 100 cm (Pre-1800–1970) | C, Hg, N, P |
| *Robbins & Edgington (1974)* | Southern Lake Michigan | 40 cm (1969) | Pb |
| *Robbins, Edgington & Parker (1974)* | Lake Michigan | 15 cm (1884–1974) | SiO$_2$ |
| *Sridharan & Lee (1974)* | Lower Green Bay, Lake Michigan | 40 cm (1969) | Al, Ca, Fe, P |
| *Walters, Kovacik & Herdendorf (1974)* | Western Lake Erie | 60 cm (1972) | Hg |
| *Walters & Wolery (1974)* | Lake Erie | 60 cm (1973) | Cr, Hg, Ni |
| *Walters, Wolery & Myser (1974)* | Lake Erie | 120 cm (1972) | As, Cd, Co, Cr, Cu, Fe, Hg, Ni, Sb, Zn |
| *Wolery & Walters (1974)* | Western Lake Erie | 50 cm (1972) | Hg |
| *Fitchko & Hutchinson (1975)* | River mouths around the Great Lakes | 60 cm (1973) | Ag, Cd, Co, Cr, Cu, Hg, Mn, Ni, Pb, Zn |
| *Petersen (1975)* | Lake Erie | 500 cm (~7,500 yr BP–1967) | Ca, Mg, P |
| *Robbins & Callender (1975)* | Lake Michigan | 350 cm (1972) | Mn |
| *Edgington & Robbins (1976)* | Southern Lake Michigan | 18 cm (1972) | Pb |
| *Förstner (1976)* | Lakes Erie and Michigan | Review Article | |
| *Kemp & Thomas (1976a)* | Lakes Ontario, Erie, and Huron | 140 cm (1970) | Al, Be, Ca, Cd, Cu, Fe, K, Mg, Mn, Na, P, Pb, S, Si, Ti, V, Zn |
| *Kemp & Thomas (1976b)* | Lakes Ontario, Erie, and Huron | 140 cm (1970) | Al, Be, Ca, Cd, Cu, Fe, K, Mg, Mn, Na, P, Pb, S, Si, Ti, V, Zn |
| *Kemp et al. (1976)* | Lake Erie | 140 cm (1970) | Al, Be, Ca, Cd, Cu, Fe, K, Mg, Mn, Na, P, Pb, S, Si, Ti, V, Zn |
| *Robbins & Edgington (1976)* | Southern Lake Michigan | 24 cm (1800–1972) | As, Ba, Br, Ca, Ce, Co, Cr, Cs, Cu, Eu, Fe, Gf, K, La, Lu, Mg, Mn, Na, Ni, Pb, Rb, Sb, Se, Sm, Tb, Th, U, Zn |

(Continued)

| References | Location | Max core length (time period) | Analytes |
|---|---|---|---|
| *Torrey (1976)* | Lake Michigan | Review Article | |
| *Williams, Murphy & Mayer (1976)* | Lake Erie | 20 cm (1971) | P |
| *Mothersill (1977)* | Thunder Bay, Lake Superior | 35 cm (1974) | Co, Cr, Cu, Fe, Mn, Ni, Zn |
| *Nriagu (1978)* | Lakes Ontario, Superior, and Erie | 30 cm (1975) | SiO$_2$ |
| *Kemp et al. (1978)* | Lakes Superior and Huron | 75 cm (1970) | Al, Be, Ca, Cd, Cu, Fe, K, Mg, Mn, Na, P, Pb, S, Si, Ti, V, Zn |
| *Warwick (1978)* | Bay of Quinte, Lake Ontario | 2 m (874 B.C.–1972) | P |
| *Nriagu et al. (1979)* | Lake Erie | 30 cm (1976) | Cd, Cu, Pb, Zn |
| *Robbins (1980)* | Southern Lake Huron | 50 cm (1975) | Al, As, Ba, Br, Ca, Cd, Ce, Co, Cr, Cs, Cu, Eu, Fe, K, La, Lu, Hg, Hf, Mg, Mo, Mn, Na, Ni, P, Pb, Rb, Sb, Sc, Si, Sm, Sn, Sr, Ti, Th, U, V, Yb, Zn |
| *Christensen & Chien (1981)* | Green Bay and Northern Lake Michigan | 15 cm (1751–1978) | As, Cd, Pb, Zn |
| *Goldberg et al. (1981)* | Lake Michigan | 60 cm (1830–1978) | Al, Cd, Co, Cr, Cu, Fe, Mn, Ni, Pb, Sn, V, Zn |
| *Manning, Lum & Birchall (1983)* | Lake Ontario | 15 cm (1820–1981) | Al, Ca, Cd, Co, Cr, Cu, Fe, Mn, Ni, P, Pb, Zn |
| *Nriagu, Wong & Snodgrass (1983)* | Toronto and Hamilton Harbors | 100 cm (1981) | Cd, Cr, Cu, Fe, Mn, Ni, Pb, Zn |
| *Breteler et al. (1984)* | Lake Ontario, Lake Erie, Niagara River | 66 cm (1934–1979) | Hg |
| *Schelske, Conley & Warwick (1985)* | Bay of Quinte (Lake Ontario) | 164 cm (874 BC–1972) | P |
| *Schelske et al. (1988)* | Rochester Basin (Lake Ontario) | 150 cm (1981) | P |
| *Christensen & Osuna (1989)* | Lake Michigan | 84.0 cm (1984) | Cd, Pb, Zn |
| *McKee et al. (1989)* | Caribou sub-basin of Lake Superior | 32 cm (1986) | Cu, Mn, Pb |
| *Schelske (1991)* | Rochester Basin (Lake Ontario) | 150 cm (1981) | P |
| *Sly (1991)* | Lake Ontario | 25 cm (1750–1972) | P |
| *Mudroch (1993)* | Lake Ontario | 15 cm (1989) | Al, As, Ca, Co, Cr, Cu, Fe, K, Mg, Mn, Na, Ni, P, Pb, Si, Ti, Zn |
| *Kerfoot, Lauster & Robbins (1994)* | Lake Superior | 40 cm (1850–1991) | Cu |
| *Mayer & Johnson (1994)* | Hamilton Harbor | 60 cm (1898–1987) | Cd, Cu, Fe, Mn, Ni, P, Pb, Zn |
| *Ritson et al. (1994)* | Lake Erie | 60 cm (1926–1985) | Pb |
| *Graney et al. (1995)* | Great Lakes basin | 170 cm (1798–1992) | Pb |
| *Rossmann (1995)* | Saginaw Bay | 65 cm (1820–1988) | Cd, Cr, Cu, Hg, Ni, Pb, Zn |
| *Azcue, Rosa & Mudroch (1996)* | Lake Erie | 46 cm (1819–1994) | Al, As, Ca, Cd, Cr, Cu, Fe, K, Mg, Mn, Na, Ni, Pb, Si, Sr, Ti, Zn |
| *Kolak et al. (1998)* | Lakes Michigan, Ontario, and Superior | 70 cm (1800–1992) | Cu |
| *Pirrone et al. (1998)* | Lakes Ontario, Michigan, Erie | 70 cm (1800–1988) | Hg |
| *Kerfoot et al. (1999)* | Lake Superior | 50 cm (1983) | Ag, Cu, Hg, Zn |
| *Kerfoot & Robbins (1999)* | Keweenaw Waterway | 60 cm (1991) | Al, As, Ba, Ca, Ce, Co, Cr, Cs, Cu, Fe, K, Mn, Na, Zn |
| *Kolak et al. (1999)* | Lake Superior | 70 cm (1994) | Cu, Zn |

| References | Location | Max core length (time period) | Analytes |
|---|---|---|---|
| *Painter et al. (2001)* | Lake Erie | 140 cm (1997) | Al, As, Cd, Cr, Cu, Fe, Hg, Mn, N, Ni, P, Pb, Zn |
| *Jeong & McDowell (2003)* | Copper Harbor (Lake Superior) | 20 cm (2000) | Cu |
| *Marvin et al. (2003)* | Lake Ontario | 60 cm (1998) | Al, As, Cd, Cr, Cu, Fe, Hg, Mn, N, Ni, P, Pb, Zn |
| *Rolfhus et al. (2003)* | Lake Superior | 10 cm (2000 | Hg |
| *Belzile et al. (2004)* | Killarney Park, Ontario | 35 cm (2004) | As. Cd, Cu, Co. Fe, Mn, Ni, Pb, Zn |
| *Kerfoot et al. (2004)* | Lake Superior | 60 cm (1983) | Ag, Au, Cu, Hg, Zn |
| *Marvin et al. (2004)* | Great Lakes | 60 cm (1840–2000) | Hg |
| *Reavie et al. (2005)* | Lake George (Ontario, Canada) | 50 cm (1770–1993) | Al, As, Br, Ca, Ce, Co, Cr, Cs, Eu, Fe, Hf, La, Lu, Mn, Na, Rb, Sb, Sc, Sm, Ta, Th, Ti, V, Yb, Zn |
| *Schelske, Stoermer & Kenney (2006)* | Great Lakes | 80 cm (1800–1993) | P |
| *Rossmann (2010)* | Lake Michigan | 69 cm (1884–1996) | Hg |
| *Drevnick et al. (2012)* | Great Lakes | Review Article | Hg |
| *Pompeani et al. (2013)* | Keweenaw Peninsula | 360 cm (~8800 yr BP–2012) | Fe, Mg, Pb, Ti |
| *Rossmann, Pfeiffer & Filkins (2014)* | Lake Michigan | 69 cm (1884–1996) | Pb |
| *Shaw Chraïbi et al. (2014)* | Lake Superior | 36 cm (1750–2010) | Al, Ba, Ca, Fe, K, Mg, Mn, Na, P, Si, Sr |
| *Yuan, Depew & Soltis-Muth, 2014* | Lake Erie | 40 cm (1800–2012) | Al, As, Ba, Be, Ca, Cd, Co, Cr, Cu, Fe, K, Mg, Mn, Mo, Na, Ni, Pb, Se, Sn, Ti, V, Zn |
| *Dittrich et al. (2015)* | Lake Superior | 35 cm (2012) | Fe, Mn |
| *Pompeani et al. (2015)* | McCargoe Cove, Lake Superior | 360 cm (~8800 yr BP–2012) | Cu, Fe, K, Mg, Pb, Ti |
| *Kerfoot et al. (2016)* | Keweenaw Peninsula | 80 cm (1800–2004) | Cu, Hg |
| *Yin et al. (2016)* | Lake Michigan | 1720–2013 | Hg |
| *Alsaffar (2017)* | Lake St. Clair | 142 cm (2015) | As, Ba, Cd, Cr, Cu, Fe, Hg, Mn, Ni, Pb, Zn |
| *Sgro & Reavie (2017)* | Lake Erie | 49 cm (1870–2011) | Al, Ca, CaO, Cd, Cu, Fe, K, Mg, MgO, Na, Ni, Pb, Si, SiO₂ |
| *Yuan (2017)* | Sandusky Basin, Lake Erie | 40 cm (1800–2012) | Al, Ca, Fe, K, Mg, Mn, Na, As, Ba, Be, Cd, Co, Cr, Cu, Mo, Ni, Pb, Se, Sn, Ti, V, Zn |
| *Sgro & Reavie (2018)* | Lake Huron | 36 cm (1823–2012) | Al, Al₂O₃, As, Ba, BaO, Ca, CaO, Cd, Co, Cr, Cs, Cu, Dy, Er, Eu, Fe, Fe₂O₃, Ga, Gd, Ho, K, K₂O, La, Li, Lu, Mg, MgO, Mn, MnO, Mo, Na, Na₂O, Nb, Nd, Ni, P, P₂O₅, Pb, Pr, Rb, Sb, Sc, Si, SiO₂, Sm, Sn, SrO, Ta, Tb, Th, TiO₂, Tm, U, V, Y, Yb, Zn |

chromite and nickel ores. Overlying these beds are mafic and felsic (volcanic rocks), often with ores of gold, silver, copper and zinc. At the top of the sequence are sedimentary rocks which yield manganese, barite and iron ores" (*Kerfoot & Nriagu, 1999*). These greenstone belts are up to hundreds of kilometers long and wide.

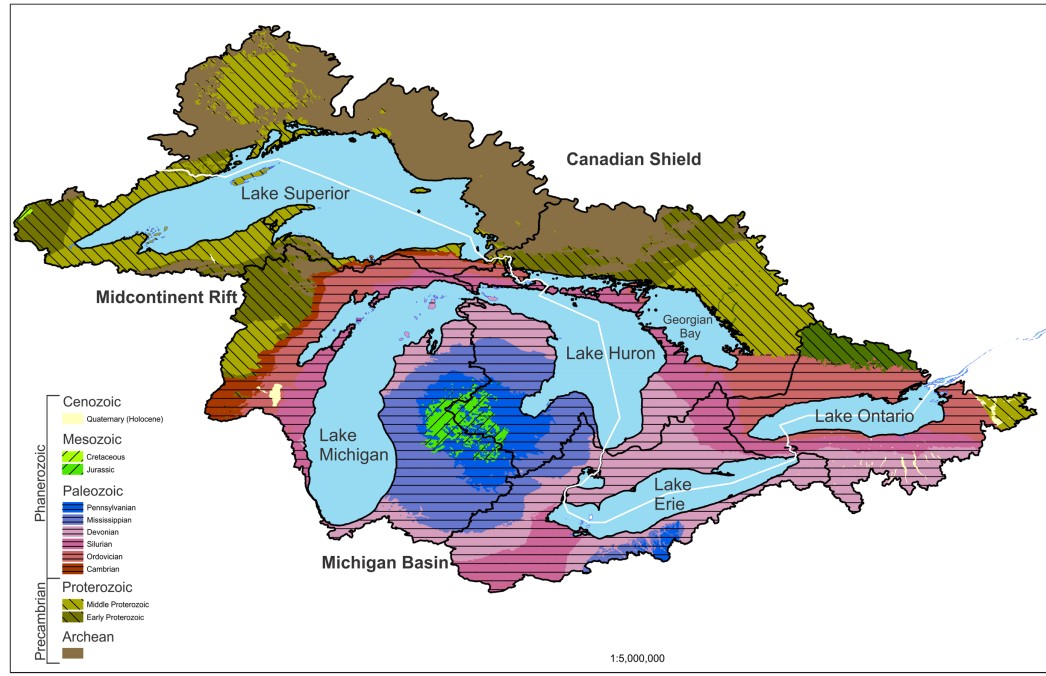

**Figure 1 Geological map of the Great Lakes watershed.** Geology of the region is summarized by geological age which is matched to rock types in the description. The basin is bounded to the north by the upland Precambrian-age Canadian Shield. The basin is bound to the south by a lowland region of Paleozoic sedimentary rocks. The Midcontinent Rift System makes up the western Lake Superior basin. United States data are from the USGS (*U.S. Geological Survey, 2018*) and state geological surveys. Canadian data are taken from GeologyOntario (*Ontario Ministry of Energy, Northern Development & Mines, 2019*).

The Midcontinent Rift System in the western Lake Superior basin is a package of basalt flows and sedimentary rocks up to 25 km thick (*Keays & Lightfoot, 2015*). Between approximately 1.109 and 1.084 billion years ago (Early to Middle Proterozoic, Fig. 1), there was a prolonged period of extension in the middle of what is now North America, which produced large quantities of basaltic magmas and a large number of sills, dikes, mafic intrusive complexes, and intrusions of alkali and carbonatitic composition (*Keays & Lightfoot, 2015*; *Kerfoot & Nriagu, 1999*). Most of the exposed rift outcrop is found around Lake Superior, but evidence suggests that it also underlies the Michigan Basin feature of Lower Michigan (Fig. 1). The rift structure extends to the southwest as far as Kansas. Many of the major metallic mineral prospects and ore bodies in the Great Lakes region are part or product of the Midcontinent Rift (*Ojakangas, Morey & Green, 2001*).

The Lake Huron basin is largely within the Paleozoic sedimentary rock region, though the northern shore is on the edge of the Canadian Shield (*Hough, 1958*). Lakes Michigan, Ontario and Erie are almost all entirely within the Paleozoic province, which is dominated by the sedimentary rocks of the Michigan Basin (Fig. 1). Surrounding the Michigan Basin is an erosion-resistant sequence of Silurian dolomite. On top of this dolomite sandstone, shale, limestone, and evaporite sediments were deposited (*LaBerge, 1994*). Lake Erie is largely underlain and surrounded by Devonian limestones and shales, and Lake Ontario is underlain by Ordovician limestones and shales. The Silurian rocks outcrop between the

**Table 2 Range of As concentrations in ppm in different rock types.** Modified and combined version of two tables presented and included in a study by the *National Research Council (US) (1977)*.

Arsenic concentration, ppm

|  | No. Analyses | Range usually reported | Average |
|---|---|---|---|
| Igneous Rocks |  |  |  |
| Ultrabasic | 37 | 0.3–16 | 3.0 |
| Basalts, gabbros | 146 | 0.06–113 | 2.0 |
| Andesites, dacites | 41 | 0.5–5.8 | 2.0 |
| Granitic | 73 | 0.2–13.8 | 1.5 |
| Silicic volcanic | 52 | 0.2–12.2 | 3.0 |
| Sedimentary Rocks |  |  |  |
| Limestones | 37 | 0.1–20 | 1.7 |
| Sandstones | 11 | 0.6–120 | 2.0 |
| Shales and clays | 324 | 0.3–490 | 14.5[a] |
| Phosphorites | 282 | 0.4–188 | 22.6 |
| Sedimentary iron ores | 110 | 1–2,900 | 400? |
| Sedimentary manganese ores | — | (up to 1.5%) | — |
| Coal | 1,150 | 0–2,000 | 13[b] |

Notes:
[a] Excluding one sample with arsenic at 490 ppm.
[b] *Boyle & Jonasson (1973)* gave 4 ppm.
Estimated on the basis of data of *Onishi (1969)* and *Boyle & Jonasson (1973)*.

two lakes, forming the Niagara Escarpment, which is the foundation of Niagara Falls (*Sly & Thomas, 1974*; Fig. 1).

The overlying sediment in the Great Lakes basin is strongly influenced by glaciation. The lowland region which includes the Erie and Michigan basins along with most of Huron and Ontario is blanketed by glacial sediments often between 50 and 350 m in thickness. The upland region of Superior, Georgian Bay, and Ontario basins has thin discontinuous layers of glacial sediments (*Larson & Schaetzl, 2001*). For a more detailed description of glacial processes in the Great Lakes, see *Larson & Schaetzl (2001)* and *Sly & Thomas (1974)*. Because sediments may be derived from glacial till transported long distances, sourcing of materials is difficult without more in-depth mineralogical analyses. Hence, excepting documented and localized metallic ores, confirming ultimate sources for materials found in lake sediment cores can be challenging.

## Metals in the Great Lakes

A discussion of the influence that rock type can have on elemental concentrations can provide some geological context for the presence of certain metallic analytes in lake sediments. For example, arsenic (As) generally occurs in lower concentrations in igneous rocks compared to sedimentary rocks, as shown in Table 2 which is a combined version of two tables from *National Research Council (US) (1977)*. The range of metal ppm values can be quite broad, even within a specific rock type. For instance, the Biwabik Iron Formation in Minnesota is a sedimentary iron ore deposit, and the range of its As content as reported by *Morey (1992)* was <3–40 ppm which is small relative to the 1 to 2,900 ppm range shown

in the table. Other metals show different ranges of composition based on rock type, but it is beyond the scope of this report to provide those details for all metals.

These bedrock characteristics can provide some information on sources of metals in the environment, but what best explains an element's presence in Great Lakes sediment cores, especially during the Anthropocene epoch? While bedrock geology is important to consider, unconsolidated surficial materials are the more significant contributors to sediment loading. As reported to the U.S. Army Corps of Engineers by the *Great Lakes Commission (2008)*, more than 65 million tons of soil may be eroding annually in the Great Lakes basin based on modeling data by *Ouyang & Bartholic (2003)*, with agricultural land being the largest known contributor to soil erosion in the basin (65–77%). Therefore, soils are probably the most relevant source of catchment inputs, but with the understanding that soils are also derived from bedrock sources that may not be proximal to eroding soils, they may not reflect local bedrock geology. A large portion of lake contamination is also sourced atmospherically, such as by fossil fuel consumption and metal smelting (refining), making the identification of a specific contributor even more difficult. What this review shows is that human activity and society's reliance on and demand for natural resources is ultimately the driver for nearly every metal or metal-containing compound whose extraction, production, and use contribute to their presence in the environment above background levels.

What follows is an encyclopedic list of metals in Great Lakes sediments. All available research, and best judgement, was used to characterize metals as they pertain to Great Lakes geochemistry. This included as necessary for each analyte: source, known history, and contaminant properties.

### Alkali metals

*Lithium*

*Contaminant status in the Great Lakes*: Unknown but possible.

*Dominant sources*: Weathering of rocks.

*Background and relevance to the Great Lakes:* Lithium (Li) is the 27th most abundant element in the Earth's crust and has been found to have a concentration of 0.07–40 µg/L in fresh waters (*Shahzad et al., 2016*). Li is found in small amounts in nearly all rocks, averaging 40 ppm in soil (*Emsley, 2002*). At low concentrations, it seems to stimulate plant growth (*Aral & Vecchio-Sadus, 2008*). Lithium's biological impacts are not well understood, although it is commonly used as a psychoactive drug. Li is an issue of emerging concern in surface waters due to its increasing use in drugs, batteries, and alloys (*Tkatcheva et al., 2015*). However, its prevalence is not well known because environmental releases of Li are not regulated in North America. It can be used in paleolimnology to account for increasing sediment inputs into a system in order to determine what amount of trace metal inputs are from anthropogenic sources in addition to natural sediment inputs (*Ikem & Adisa, 2011*). Historically, increases in Li in the sediment record of Northeast Germany indicated erosion of the catchment area (*Selig, Leipe & Dörfler, 2007*). *Rossmann & Barres (1988)* found median Li concentrations (dissolved and particulate) ranged from 0.64 ppb in Lake Superior waters to 2.4 ppb in Lake Ontario in samples collected between

1981 and 1983. They found that Lakes Erie, Michigan, and Ontario had significantly ($p >$ 0.05) higher concentrations of Li than Lake Superior waters. In Lake Superior, *Nussmann (1965)* found Li concentration did not vary systematically in sediment cores. Concentrations in surface sediments averaged 45 ppm, while the average concentration in glacial sediments was 96 ppm. *Nussmann (1965)* theorized that the fact Li was not sorbed in clays may mean there is little sediment-water exchange of Li in Lake Superior.

*Sodium*

*Contaminant status in the Great Lakes:* Associated with urban environments, mining and other industries.

*Dominant sources:* The Michigan basin (Fig. 1) due to evaporation of sea water, road salt applications, domestic and industrial wastewater.

*Background and relevance to the Great Lakes:* Sodium (Na) is the 6th most abundant element, making up 2.36% of the Earth's crust (*Lindsay, 1968*; *Taylor, 1964*). Na is an essential element to life as the principal cation of extracellular fluids in higher animals. It is readily weathered and highly soluble in aqueous environments, where it has a long residence time (*Mittlefehldt, 1999b*). Na enters the hydrosphere from urban environments through road salt and waste water treatment plants (*Chambers et al., 2016*). Due to its long residence time, Na has been persistently higher than background levels in Lakes Superior, Michigan, and Huron over the last 150 years, though it did peak in Lakes Erie and Ontario between 1965 and 1975, decreasing in more recent years due to a reduction in point source discharges (*Chapra, Dove & Warren, 2012*). *Kemp & Thomas (1976b)* and *Kemp et al. (1978)* considered Na a conservative element in their cores from Lakes Ontario, Erie, Huron, and Superior over the period from pre-settlement to the early 1970s and observed relatively stable concentrations over the interval. *Shaw Chraïbi et al. (2014)* found increases in Na starting in 1952 with a gradual decline between 1952 and 1990 in their core from eastern Lake Superior. In their core of western Lake Superior, they found stable concentrations of Na up to 1955 ending with an increase in concentration likely resulting from urban inputs in the vicinity of Duluth, Minnesota. *Reavie et al. (2005)* found that Na concentrations gradually increased in the 1800s with European settlement in their core from Lake George, just downgradient from Lake Superior. *Sgro & Reavie (2017)* attributed Na in their cores in Lake Erie to road salt impacts and found the highest concentrations in surface sediments. In Lake Huron, *Sgro & Reavie (2018)* associated Na with edaphic deposits and found concentrations remained stable until around 2011.

*Potassium*

*Contaminant status in the Great Lakes:* Yes, related to industrial and agricultural activity.

*Dominant sources:* Weathering.

*Background and relevance to the Great Lakes:* Potassium (K) is the 7th most abundant element, making up 2.09% of the Earth's crust (*Schoeld, 1968*; *Taylor, 1964*). It is found in all soils and exists in several forms, some readily weathered and dissolved. K is an essential nutrient in both plant and animal cells, being more important in plants as Na is more important in animals. K leaches into rivers from fertilizer applications and is released with

burning biomass (*Schoeld, 1968*; *Pompeani et al., 2013*). *Kemp & Thomas (1976b)* and *Kemp et al. (1978)* considered K a conservative element in their cores from Lakes Ontario, Erie, Huron, and Superior over the period from pre-settlement to the early 1970s and observed relatively stable concentrations over the interval except near the sediment-water interface, where they observed increases and decreases in concentration related to fluctuating levels of organic matter. *Shaw Chraïbi et al. (2014)* found increases in K in 1952 associated with mining activity with a gradual decline between 1952 and 1990 in their core of eastern Lake Superior. In their core of western Lake Superior they found stable concentrations of K up to 1955, ending with an increase in concentration. *Pompeani et al. (2015)* observed increases in K concentration in their core near Isle Royale, Lake Superior around 5800 BP that matched the dating of Cu artifacts found in the region, suggesting that concentrations were related to ancient mining practices. K concentrations do not increase with modern mining techniques, implying inputs from wood fires or perhaps some other difference between modern and ancient mining (*Pompeani et al., 2015*).

*Rubidium*
*Contaminant status in the Great Lakes:* None known.
*Dominant sources:* Lithogenic, silicates, coal burning.
*Background and relevance to the Great Lakes:* Rubidium (Rb) is a widely distributed element that forms no minerals. It is concentrated in silicates and is soluble in aqueous solutions (*Simmons, 1999*). It has a concentration of about 90 ppm in the Earth's crust (*Mosheim, 1968b*; *Taylor, 1964*). Rb is considered a conservative lithogenic element and can be used to control for natural inputs when measuring the extent of anthropogenic inputs of other elements. Rb is also thought to be immobile in the sediments (*Boës et al., 2011*). It is considered an ultra-trace element for humans and other organisms, and it undergoes consistent biomagnification in the food web (*Campbell et al., 2005*). Most Rb in lakes is related to geologic features and minerals in soils. A possible anthropogenic source is coal burning, especially from the Nanticoke Generating Station on Lake Erie, which was the largest coal-fired power station in North America until it was decommissioned in 2013 (*Campbell et al., 2005*).

*Cesium*
*Contaminant status in the Great Lakes:* Yes, in the form of $^{137}$Cs.
*Dominant sources:* Erosion and weathering. Radioactive isotope $^{137}$Cs from atmospheric deposition after nuclear weapons testing.
*Background and relevance to the Great Lakes:* Cesium (Cs) is widely distributed in the Earth's crust at low concentrations (*Mosheim, 1968a*). It averages 1 ppm in granite and 4 ppm in sedimentary rocks. In fresh water, Cs occurs naturally from erosion and weathering and at between 0.01 and 1.2 µg/L. Cs has serious health impacts in the form of its radioactive isotopes, which cause tissue damage and disruption of cellular function. Cs is used in paleolimnological studies because the isotope $^{137}$Cs is a radionuclide marker for the early 1960s (*Gobeil, Tessier & Couture, 2013*). The majority of $^{137}$Cs in surface waters in North America, including the Great Lakes, is due to nuclear weapons testing, which began

in 1952 and ended in 1963 with the Limited Test Ban Treaty. In the Great Lakes, the maximum deposition of $^{137}$Cs was in the spring of 1963, though peaks in the sediment may be as much as two years later (*Agency for Toxic Substances & Disease Registry (ASTDR), 2004a*; *Reavie et al., 2005*). The radionuclide is strongly bound to expandable lattice clay minerals such as illite in the Great Lakes, so it tends to be a reliable temporal indicator in sediment profiles.

### Alkaline earth metals

*Magnesium*

*Contaminant status in the Great Lakes:* Some historical contamination associated with taconite tailings in the western part of Lake Superior.

*Dominant sources:* Limestone in Lake Michigan basin (Fig. 1); dolomite content of bluffs along north shore of Lake Erie (*Chapra, Dove & Warren, 2012*).

*Background and relevance to the Great Lakes:* Magnesium (Mg) is abundant in nature, always as part of a compound. It is the 8th most abundant crustal element and 6th most abundant metallic element, making up 2.5% of the Earth's crust (*Gross, 1968*). Mg readily dissolves during weathering of rocks and enters the hydrosphere (*Mittlefehldt, 1999a*). It is a necessary element for life and one of the four bulk metals in the human body (*Schroeder, Nason & Tipton, 1969*). In lake sediment records, increases in Mg are frequently associated with erosion (*Mackereth, 1966*; *Norton et al., 1992*), and in more modern times, Mg is associated with road dust (*Spengler et al., 2011*). *Kemp & Thomas (1976b)* and *Kemp et al. (1978)* found Mg as one of the major elements in the sediment of Lakes Ontario, Erie, Huron and Superior and found it to be generally uniform in all of their cores. In a sediment core from western Lake Superior, *Shaw Chraïbi et al. (2014)* found peaks in Mg around 1972, which they associated with taconite processing and waste discharge near Silver Bay, MN, USA. *Arthur et al. (1973)* found Mg to be a component of the dissolved solids from the tailings. But considering the core locations of *Shaw Chraïbi et al. (2014)* and the circulation patterns of Lake Superior, the association in their cores is unclear. *Pompeani et al. (2015)* found Mg concentrations in Lake Superior near McCargoe Cove to be relatively stable over their 8,400 year record, though increases are believed to reflect changes in the delivery of metals to the basin, as high concentrations of Mg are found in the bedrock surrounding Cu lodes.

*Calcium*

*Contaminant status in the Great Lakes:* Yes, related to runoff and mining.

*Dominant sources:* Carbonate minerals such as those found in the Midcontinent Rift and in the Paleozoic (carbonate-rich) bedrock surrounding the lower Great Lakes (Fig. 1); detrital material.

*Background and relevance to the Great Lakes:* Calcium (Ca) is the 5th most abundant element in the Earth's crust. It is one of the most widely distributed elements, present in almost all natural waters. Ca is the most common inorganic element in higher animals and necessary for most life, concentrating in skeletons, exoskeletons, and shells and distributed throughout tissues. It is also needed for plant meiosis and the maintenance of soil pH

(*Mantell, 1968*). In urban environments, Ca pollution comes from building materials (*Chambers et al., 2016*). The Great Lakes are rich in limestone (especially in the Paleozoic province surrounding the lower Great Lakes (Fig. 1)) and therefore calcium carbonate ($CaCO_3$). In the summer, the lakes can experience periods of "whiting", when warming waters release $CaCO_3$ into the water column and dissolved $CO_2$ is removed by algae. Sediment cores in the Great Lakes have fluctuating $CaCO_3$ levels with times of warmer climate (*Meyers, 2003*). Due to its long residence time, Ca has been persistently increasing in Lakes Superior, Michigan and Huron over the last 150 years, though it did peak in Lakes Erie and Ontario between 1965 and 1975, decreasing in more recent years due partly to the introduction of quagga mussels (*Chapra, Dove & Warren, 2012*), who use the Ca in formation of shells. In Lakes Superior and Huron, *Kemp et al. (1978)* considered Ca a conservative element, representing terrigenous sources of materials in their sediment cores. In Lakes Ontario and Huron, *Kemp & Thomas (1976b)* grouped Ca with the carbonates, finding decreasing concentration in parts of Lake Erie and increases to very high surface concentrations in Lake Ontario above the *Ambrosia* pollen horizon (marking European settlement and deforestation), which they attributed to the dissolution of detritus. *Shaw Chraïbi et al. (2014)* found increases in Ca in 1952 that they associated with overall mining activity in the basin, with a gradual decline between 1952 and 1990 in their core of eastern Lake Superior. In their core of western Lake Superior, they found stable concentrations of Ca until 1955 followed by an increase in concentration.

*Strontium*

*Contaminant status in the Great Lakes:* Yes, related to mining, fossil fuels, and industry.
*Dominant sources:* Ca-containing rocks and minerals, dust, mining waste, fossil fuels, other industries.
*Background and relevance to the Great Lakes:* Strontium (Sr) is the 15th most common element on Earth, making up 0.034% of the Earth's crust (*Höllriegl & München, 2011*). Mean dissolved Sr in the Great Lakes ranged from 42 ppb in Lake Superior to 180 ppb in Lake Ontario between 1981 and 1985 (*Rossmann & Barres, 1988*). There is a small amount of Sr in many rocks and minerals, but it is present in all Ca-bearing rocks, minerals, and soils. Sr is naturally released from the Earth's crust as dust and wind. Human processes that contribute to Sr release include mining, milling and processing, pyrotechnics and phosphate fertilizers. Other sources for Sr pollution include association with chloride in road salt in the Lake Ontario watershed (*Meriano, Eyles & Howard, 2009*) and iron corrosion products from residential drinking water distribution systems (*Gerke et al., 2013*). It is also present in coal and becomes concentrated in coal fly ash (*Sherman et al., 2015*). *Spengler et al. (2011)* also attributed increases in Sr with road dust, brake dust, and diesel tail pipe emissions near Buffalo Peace Bridge on the Niagara River between Lakes Erie and Ontario. Sr is chemically similar to Ca and can deposit in bones. In fish, otoliths and other biogenic carbonates Sr and Sr/Ca ratios can be used to reconstruct past water temperatures and salinity because the partitioning constant for those elements varies with those parameters (*Limburg et al., 2015*; *Carilli et al., 2015*). Sr found at natural

concentrations has low toxicity, but radioactive Sr can be dangerous, especially because it deposits in bones, leading to bone cancer (*Hampel, 1968e*). Sr has several radioactive isotopes, the longest lived being $^{90}$Sr with a half-life of 29 years. Radioactive Sr was released into the atmosphere between 1945 and 1980 via nuclear weapons testing. While present in Great Lakes water, a 1984 study indicated that radioactive Sr was within safe drinking water parameters (*Durham & Joshi, 1984*), and it is likely that radioactivity has since decreased. *Shaw Chraïbi et al. (2014)* found increases of Sr from pre-settlement 1952 in their eastern core of Lake Superior associated with mining. In their western core, they found a peak of Sr between 1955 and 1967, which they associated with taconite processing in Silver Bay.

*Barium*

*Contaminant status in the Great Lakes:* Yes, listed as a sediment and water contaminant of the Saginaw Bay and the Rouge River Great Lakes Areas of Concern (AOCs) (*Hartig & Thomas, 1988*).

*Dominant sources:* Barite, drilling fluid (oil and gas industry).

*Background and relevance to the Great Lakes:* Barium (Ba) is the 14th most abundant element in the Earth's crust (*Hampel, 1968a*; *Smith, 1999*). Ba is highly reactive and is precipitated with sulfate and carbonates. Therefore, it is not very mobile in soils and remains bound to particles in circumneutral pH (*Madejón, 2013*). When soluble, Ba is moderately acutely toxic (*Moore, 1991*). Barite (barium sulfate) is used as a drilling fluid/mud for oil and gas wells. Ba was historically used as a tracer in place of lead for vehicle emissions via diesel and unleaded gasoline (e.g., in sediment cores from an urban Hawaii stream described by *Sutherland (2000)*), but newer diesel fuels are low-sulfur and do not need to use Ba as a sulfur-scavenging agent (*Agency for Toxic Substances & Disease Registry (ATSDR), 2007b*). Ba was found in at least 798 of 1,684 current or former EPA National Priorities List (NPL) hazardous sites in the United States. In Lake St. Clair, sediment Ba was at non-detect levels in 1983 and up to 48.37 mg/kg in 2010 (*Alsaffar, 2017*). According to USEPA guidelines, 20–60 mg/kg is considered moderately polluted with Ba, and >60 mg/kg is heavily polluted. In Saginaw Bay, Ba levels up to 422 mg/kg were found (*Hartig & Thomas, 1988*). In contrast, Ba was one of the elements least enriched by anthropogenic sources in Lake Erie according to *Yuan (2017)*. In Lake Superior, *Shaw Chraïbi et al. (2014)* saw, in an eastern core, an increase in Ba from post-settlement through ~1,952 associated with mining, peaking around 1960. In a western core, they observed a Ba peak between 1955 and 1967 associated with taconite processing waste discharge near Silver Bay.

**Transition metals**

*Scandium*

*Contaminant status in the Great Lakes:* None known.

*Dominant sources:* Widely distributed.

*Background and relevance to the Great Lakes:* Scandium (Sc) is the 36th most abundant element in the Earth's crust (*Horovitz, 1975*). Sc is present in small amounts in

hundreds of minerals, as there are no geological processes to concentrate it and it lacks affinity for common ore-forming ions (*Emsley, 2014*; *U.S. Department of the Interior (USDOI) & U.S. Geological Survey (USGS), 2017*). Hence, no localized sources of Sc are known in the Great Lakes basin, and no Sc is mined in the United States. Sc is not known to be a nutrient for any organism and has, in fact, been found to be toxic in high concentrations due to its high affinity for proteins (*Sánchez-González et al., 2013*). According to *Horovitz (1975)*, the average soil content of Sc is 7 ppm, with a range of 3–50 ppm. Sc is a useful reference element that can be compared to the concentration of other, less conservative elements in order to gain more useful information about that element's variability in time (*Peirson, Cawse & Cambray, 1974*; *Dias & Prudêncio, 2008*). Sc has not been greatly studied in the Great Lakes, although *Winchester (1972)* found the natural inputs of Sc to Lake Michigan to be 0.004 ppb per year and the natural output to be 0.027 ppb per year. *Shimp et al. (1971)* and *Reavie et al. (2005)* found that Sc concentrations did not vary over time in their sediment cores from southern Lake Michigan and Lake George, respectively.

### Titanium

*Contaminant status in the Great Lakes:* Some atmospheric contamination from vehicle and coal emissions along with catchment erosion.
*Dominant sources:* Ubiquitous.
*Background and relevance to the Great Lakes:* Comprising 0.63% of the Earth's crust by mass, Titanium (Ti) is the 9th most abundant element and the 7th most abundant metal. It is widely distributed in a diverse range of rocks, soils, minerals and atmospheric dust (*Barksdale, 1968*). According to *Woolrich (1973)*, Ti ranges in concentration in the air from 0.003 to 0.07 ppb. It is considered a weathering-resistant, immobile mineral, generally insoluble and not mobilized by acidification (*Norton et al., 1992*; *Schaetzl, Mikesell & Velbel, 2006*). Increases in Ti are generally associated with erosion. Titanium dioxide ($TiO^2$) is useful as an indicator of erosion rates and for the normalization of other trace metals, but one must take care to account for atmospheric concentrations of Ti, which may lead to inconsistent results (*Norton et al., 1992*). *Spengler et al. (2011)* performed a study of air toxics near the Buffalo Peace Bridge U.S. Border crossing on the east end of Lake Erie and found two-fold increases in Ti downwind of the city of Buffalo, which they attributed to coal combustion, traffic emissions, and road dust. In rock, Ti is known to occur in gabbroid rocks in the Midcontinent Rift near the south shore of Lake Superior (Fig. 1; *Wisconsin Geological & Natural History Survey, 2019*), but it is probably much more widespread. Ti is considered a conservative element and was found to be generally constant in sediment cores from Lakes Ontario, Erie, Huron and Superior (*Kemp et al., 1976*, *1978*; *Pompeani et al., 2015*).

### Vanadium

*Contaminant status in the Great Lakes:* Yes, especially in Lake Michigan due to past contamination from the BP Whiting Refinery.

*Dominant sources:* Weathering, fossil fuel emissions.

*Background and relevance to the Great Lakes:* Vanadium (V) is the 22nd most abundant element in the Earth's crust at around 130 ppm by mass (*Foote Mineral Co., 1968*; *Sigel, Sigel & Sigel, 2013*). V is non-toxic at common concentrations, though it can bioaccumulate, and at higher concentrations it can have negative effects on the respiratory system. It is an essential trace element for many organisms, and deficiency is associated with a variety of conditions including problems in growth, reproduction, and blood formation (*Gummow, 2011*; *Venkataraman & Sudha, 2005*). V is present in more than 50 minerals and its main natural source in the environment is weathering of rocks, hence it is difficult to determine specific rock sources in the Great Lakes. Anthropogenic inputs are largely atmospheric; V is a major component of coal-powered and other fossil fuel emissions. V has been a contaminant of more recent concern and has been found in 319 out of 1,699 hazardous sites on the EPA's National Priorities List (*Agency for Toxic Substances & Disease Registry (ATSDR), 2012c*). *Shahin et al. (2000)* found an increase of V dry deposition in the urban area of Chicago compared to other sampling sites around Lake Michigan due to emissions from industrial sources. *Kemp et al. (1978)* found V to be stratigraphically uniform in most of their cores of Lakes Huron and Superior taken in the early 1970s, though there were anthropogenic inputs. One of the major sources of V pollution on the Great Lakes is the BP Whiting Refinery on the southwestern shore of Lake Michigan. This refinery stored petroleum coke (petcoke), a byproduct of the refining process, at a site in Chicago operated by KCBX Terminals Company until 2016. The mounds of petcoke at this site were a source of windborne V and negatively impacted air quality in the area (*Agency for Toxic Substances & Disease Registry (ASTDR), 2016*). BP also discharged 39 barrels of oil into Lake Michigan in March 2014, and the EPA found that they failed to implement a proper spill prevention plan (*Bassler, 2016*).

*Chromium*

*Contaminant status in the Great Lakes:* Yes, listed as a sediment contaminant or water contaminant in 17 AOCs in the Great Lakes (*Hartig & Thomas, 1988*).

*Dominant sources:* Greenstone belts in Archean rocks in Canadian Shield north of Lake Superior (Fig. 1), chromine, steel production, leather tanning.

*Background and relevance to the Great Lakes:* Chromium (Cr) is an essential human nutrient at low concentrations but is toxic at higher levels, acting on the respiratory tract and as a carcinogen (*Ilton, 1999*; *Agency for Toxic Substances & Disease Registry (ATSDR), 2012b*). Major anthropogenic sources of Cr in natural waters include waste from electroplating, leather tanning, and textile industries. Cr is important in use of the steel industry, as it is an essential component of stainless steel (*U.S. Department of the Interior (USDOI) & U.S. Geological Survey (USGS), 2017*). *Yuan (2017)* included Cr in group of elements most enriched by anthropogenic loading in Lake Erie. *Walters, Wolery & Myser (1974)* also observed significant enrichment in Lake Erie cores since about 1950. *Reavie et al. (2005)* associated peaks of Cr in their cores of Lake George with the now defunct Northwestern Leather Company which discharged directly into St. Mary's River, just downstream from Lake Superior. Cr in their cores peaked around 1946 and sharply

dropped in the late 1950s after the tannery closed. *Kemp et al. (1978)* found only small amounts of localized enrichment in their cores of Lakes Superior and Huron.

*Manganese*

*Contaminant status in the Great Lakes:* Yes, listed as a sediment contaminant or water contaminant in of the following AOCs: Clinton River, Cuyahoga River, Oswego River (*Hartig & Thomas, 1988*).

*Dominant sources:* Terrigenous sources including sedimentary rocks overlying the Canadian Shield above Lake Superior (Fig. 1), contaminant from iron and steel production; ferromanganese nodules.

*Background and relevance to the Great Lakes:* Manganese (Mn) is the 12th most abundant element in the Earth's crust and biosphere and is considered one of the five essential trace elements to nearly all organisms (*Hampel, 1968d*; *Nádáská, Lesný & Michalík, 2012*; *Uren, 2013*). In high chronic doses Mn impacts the human central nervous system (*Nádáská, Lesný & Michalík, 2012*). Mn is common in the Great Lakes, especially in Lakes Michigan and Ontario where it occurs in nodular concretions and coatings on sand particles and rocks (*Sly & Thomas, 1974*). Mn preferentially sorbs to heavy metals and nodules can act a sink for those metals; it is especially associated with Co and Fe (*Kemp et al., 1978*; *Uren, 2013*), and it diffuses and precipitates with other redox sensitive elements such as Fe and As (*Sly & Thomas, 1974*; *Azcue, Rosa & Mudroch, 1996*); hence, Mn is a common co-contaminant from mining activities. Ferromanganese concretions found in Lake Superior are thought to be linked to activity of iron oxidizing bacteria (*Dittrich et al., 2015*). Mn occurs in sediment cores but often concentrates near the sediment water interface because it is extremely redox sensitive. It tends to migrate through anoxic porewater and then becomes trapped by oxic surface sediments, thereby making it difficult to establish its actual depositional history (*Kemp & Thomas, 1976a*; *Kemp & Thomas, 1976b*; *Sly & Thomas, 1974*; *Kemp et al., 1978*; *Azcue, Rosa & Mudroch, 1996*; *Reavie et al., 2005*; *Shaw Chraïbi et al., 2014*; *Kuzyk, Macdonald & Johannessen, 2015*). In Lake Erie and shallow areas of Lake Ontario, which are seasonally anoxic, Mn is able to escape from the sediment and does not accumulate at the surface as it does in other lakes (*Kemp & Thomas, 1976b*). Peaks in Mn in lower layers of sediment cores are often related to anthropogenic activities such as taconite processing or as a contaminant from iron and steel production (*Kemp et al., 1978*; *Reavie et al., 2005*; *Chambers et al., 2016*), hence the record of Mn is particularly relevant to Lake Superior.

*Iron*

*Contaminant status in the Great Lakes:* Yes, listed as a sediment contaminant or water contaminant in 16 AOCs in the Great Lakes (*Hartig & Thomas, 1988*).

*Dominant sources:* Terrigenous sources including sedimentary rocks overlying the Canadian Shield above Lake Superior (Fig. 1), weathering, mining.

*Background and relevance to the Great Lakes:* Iron (Fe) is the 4th most abundant element in the Earth's crust (~7% or 70,000 ppm) and is believed to be the most abundant element

on Earth. Fe is essential to the biological function of all animals (*Hampel, 1968c*; *Williamson, 1999b*). Some Fe bearing minerals are the result of *Thiobacillus* reactions, which oxidize Fe in aqueous solution. Fe can scavenge trace metals and bring them out of solution, but it can also carry those same trace metals back into solution under acidic or reducing conditions. Increased Fe levels in surface waters can increase algal productivity, especially of cyanobacteria which have higher Fe requirements than eukaryotic algae. Fe becomes accessible to algae in anoxic sediments, which transform it into $Fe^{2+}$, an ion suitable for algal uptake (*Verschoor et al., 2017*). In some cases in the Great Lakes, Fe has the potential to be a limiting nutrient in the formation of algal blooms (*North et al., 2007*). The most important Fe-producing district in the United States is the Mesabi Range within the Lake Superior Watershed, where there was intense taconite processing during the 1960s and 1970s. The Reserve Mining Company in Silver Bay, Minnesota discharged taconite tailings directly into Lake Superior. In sediment cores Fe, like Mn, is a mobile element under reducing conditions which can concentrate at the sediment-water interface. *Shaw Chraïbi et al. (2014)* found increasing concentrations in their core from the eastern watershed of Lake Superior between 1850 and 1890. In the western watershed they found increases around 1880, with a peak around 1930. *Kemp & Thomas (1976b)* found Fe levels in their sediment cores from Lakes Ontario, Erie, and Huron remained fairly stable except in Lake Erie, which showed increases near the surface, possibly due to anthropogenic inputs from steel plants of Detroit and Cleveland. *Yuan (2017)* also found surface enrichments in their sediment cores from the Sandusky Bay of Lake Erie, likely due to a combination of anthropogenic loading and redox driven migration of Fe near the sediment surface.

*Cobalt*

*Contaminant status in the Great Lakes:* Yes, from mining, fossil fuel combustion, and other industrial sources.
*Dominant sources:* Natural bedrock sources, mine tailings, industry, fossil fuels.
*Background and relevance to the Great Lakes:* Cobalt (Co) has an average crustal concentration of 25 ppm (*Taylor, 1964*). Atmospheric releases of Co are expected to be mostly from natural sources such as windblown soil, forest fires, and volcanoes. Increased Co levels in water may be attributed to anthropogenic sources including mining, the burning of fossil fuels, vehicular exhaust, and the use of phosphate fertilizers (*Agency for Toxic Substances & Disease Registry (ASTDR), 2004b*). According to *Smith & Carson (1981)* Lake Huron received 76% of its Co from natural sources and Lake Superior received 85.4%. Concentrations of Co are increased in mine tailings of the Keweenaw Peninsula (Lake Superior) and are also waste contaminants related to the production of alloys and metal coating processes (*Jeong & McDowell, 2003*; *Reavie et al., 2005*). It is a redox sensitive element much like Mn and Fe and can migrate to the top of sediment cores (*Yuan, 2017*). *Kemp et al. (1978)* found uniform concentrations of Co in their cores from Lake Superior and Lake Huron with minor anthropogenic enrichments at some sites.

*Nickel*

*Contaminant status in the Great Lakes:* Yes, listed as a sediment contaminant or water contaminant in 16 AOCs in the Great Lakes (*Hartig & Thomas, 1988*).

*Dominant sources:* Greenstone belts in the Archean rocks north of Lake Superior (Fig. 1), steel mills, coal-fired power plants, mining, and other industries.

*Background and relevance to the Great Lakes:* Nickel (Ni) is the 24th most abundant element in the Earth's crust and is more abundant than Cu, Zn and Pb combined. There are naturally low levels of Ni in surface water (*Agency for Toxic Substances & Disease Registry (ASTDR), 2005a*), and it has toxicity only related to industrial exposure (*Adamec & Springer, 1968*). There are respiratory, immune system, and birth issues linked to workers in refineries (*Agency for Toxic Substances & Disease Registry (ASTDR), 2005a*). *Cole et al. (1990)* found increased Ni in a peat core in Northern Indiana, downwind of Chicago, related to steel mills and coal-fired plants. *Kemp et al. (1978)* found high anthropogenic inputs of Ni in Lake Huron related to contamination from the Ni mining complex in the Sudbury area. This contamination did not extend to Lake Superior, and the authors theorized that this is due to minimal atmospheric contamination and little migration in pore water. In Lake Superior, *Nussmann (1965)* also found low, natural Ni content in sediments that was concentrated in the clay size fraction. *Walters, Wolery & Myser (1974)* noted enrichment in their cores of Lake Erie related to electroplating facilities in the Cleveland area, particularly during the late 1940s.

*Copper*

*Contaminant status in the Great Lakes:* Yes, listed as a sediment contaminant or water contaminant in 31 AOCs in the Great Lakes (*Hartig & Thomas, 1988*).

*Dominant sources:* Precambrian Shield underlying Lake Superior (Fig. 1; *Kolak et al., 1999*), mining, wastewater, and other industries.

*Background and relevance to the Great Lakes:* Copper (Cu) is the 26th most abundant element in the Earth's crust (*Williamson, 1999a*). It is a necessary nutrient for plants, animals, and bacteria, though it is toxic at high concentrations (*Grinstead, 1968*). Because it is harmless or even beneficial at low concentrations, *Kolak et al. (1998)* consider it a noncritical contaminant, a group of anthropogenically enriched elements that are considered lower priority for contaminant loading reduction than elements that are toxic at lower concentrations. According to *Agency for Toxic Substances & Disease Registry (ASTDR) (2004c)*, waste water is the biggest anthropogenic source of Cu in waterways, and considerable amounts of Cu remain in effluent. They calculate that runoff from natural weathering and disturbed sediment accounts for 68% of Cu in natural waters. Cu has been mined for more than 6,000 years (*Peretti, 1968*). Ore deposits in the Keweenaw Peninsula (Lake Superior) were discovered in the 1840s, though evidence of Cu mining on Isle Royale goes back 6,500 years before present (*Pompeani et al., 2015*). Between 1850 and 1929 the Keweenaw district was the 2nd largest producer of Cu in the world (*Kerfoot et al., 2016*). Cu concentrations above background levels have been found in all of the Great Lakes, whether due to wastewater effluent, mining, or industry, though the levels are decreasing in more recent times, potentially due to legislation and better containment of industrial

wastes (*Kemp et al., 1976*, *1978*; *Kolak et al., 1998*). In their sediment cores of Lake Superior, *Kolak et al. (1999)* surmised Cu in the nearshore sediments were related to mining discharge and that offshore was related instead to biological uptake and settling of organic matter. *Yuan (2017)* also found Cu correlated with organic matter in their surficial sediments from Lake Erie. *Kerfoot et al. (1999)* determined less than 10% of the Cu loading into Lake Superior was atmospheric, although atmospheric deposition is a source of many other metals into the lake.

### Zinc

*Contaminant status in the Great Lakes:* Yes, listed as a sediment or water contaminant in 30 AOCs in the Great Lakes-St. Lawrence River system (*Hartig & Thomas, 1988*).

*Dominant sources:* Mafic and felsic volcanic rocks overlying the Greenstone belts in Archean rocks north of Lake Superior (Fig. 1), erosion, urban runoff, mine drainage, other industrial sources.

*Background and relevance to the Great Lakes:* Zinc (Zn) is one of the most widely used metals in the world. Though Zn is essential to human life it can be toxic at high concentrations. Zn deficiency and excess can result in problems with metabolism and organ function in humans (*Nriagu, 2011*). It has been mined extensively and is released into the atmosphere from mining, iron and steel production, and coal and fuel combustion. It is released into waterways through erosion, urban runoff, mine drainage, and municipal and industrial effluent (*Agency for Toxic Substances & Disease Registry (ASTDR), 2005b*). Sources of Zn pollution in urban environments include galvanized steel and automotive exhaust (*Chambers et al., 2016*). It has been found enriched in post-settlement surface layers of sediment cores throughout the Great Lakes, and authors agree that its sources are largely anthropogenic (*Förstner, 1976*; *Goldberg et al., 1981*; *Kemp & Thomas, 1976a*, *1976b*; *Kemp et al., 1978*; *Marvin et al., 2003*; *McKee et al., 1989*; *Nussmann, 1965*; *Painter et al., 2001*; *Reavie et al., 2005*; *Yuan, 2017*). In their cores of Lake George (between lakes Superior and Huron), *Reavie et al. (2005)* attributed Zn to the production of alloys and metal coating processes. *Goldberg et al. (1981)* concluded Zn in Lake Michigan came primarily from the iron and steel industry. *Förstner (1976)* found major increases in Zn between 1939 and 1955 related to the growth of industry during World War II and the Korean conflict.

### Yttrium

*Contaminant status in the Great Lakes:* None known.

*Dominant sources:* Minerals containing rare-earth elements.

*Background and relevance to the Great Lakes:* Yttrium (Y) is the 28th most abundant element in the Earth's crust, occurring at an average of 30 ppm. It is twice as abundant as lead. It is found in all rare earth minerals and is sometimes classified with them, although it is not technically part of the group (*Emsley, 2002*). It is very chemically similar to the rare earth elements and was formed in the Earth's crust by the same geochemical processes (*Daane, 1968*). Y has no known biological role, though it does occur in all species. It has no apparent toxicity and poses no environmental threat to plants or animals. To date, Y has

not been studied in the Great Lakes except as $^{90}$Y in water, a short-lived decay byproduct of $^{90}$Sr generated as nuclear waste (*Amiro, 1993*).

*Zirconium*

*Contaminant status in the Great Lakes:* None Known.

*Dominant sources:* Soil dust.

*Background and relevance to the Great Lakes:* Zirconium (Zr) is a lithogenic element that is immobile in the sediments and resistant to weathering (*Boës et al., 2011*; *Förstner & Wittmann, 1979*; *Schaetzl, Mikesell & Velbel, 2006*). While little studied, Zr seems to have very low toxicity (*McClain, 1968*). In sediment cores it is largely associated with soil dust, which is mostly atmospherically transported. *Nussmann (1965)* found Zr in Lake Superior was concentrated in the fine silt and was not related to biological activity or organic matter.

*Niobium*

*Contaminant status in the Great Lakes:* None known.

*Dominant sources:* Minerals containing rare-earth elements.

*Background and relevance to the Great Lakes:* Niobium (Nb) is the 33rd most abundant element in the Earth's crust (*Emsley, 2002*). It generally occurs with Ta and the rare-earth elements (*Schmidt, 1968*). It is very similar to Ta in physical, chemical, and electric properties. Niobium is used in the production of high-strength, low-alloy steel and super alloys. It is used in high-pressure pipeline construction and the auto industry (*Mackay & Simandl, 2014*). Nb has not been studied in the Great Lakes.

*Molybdenum*

*Contaminant status in the Great Lakes:* Likely.

*Dominant sources:* Steel mills and other industrial sources.

*Background and relevance to the Great Lakes:* Molybdenum (Mo) is the 54th most abundant element in the Earth's crust and averages around 2 ppm in soils (*Emsley, 2002*). It is used in steel making and is found in higher concentrations near mills, industrial sources, and uranium mines (*Alloway, 2013a*; *Giussani, 2011*). It is also found in waste water sludge. Mo is an essential trace element in all species, and it is critical for nitrogen fixation in plants. It is generally toxicologically harmless in humans (*Sigel, Sigel & Sigel, 2013*). *Shahin et al. (2000)* found significantly higher atmospheric fluxes of Mo near urban sites than more remote sites around Lake Michigan. *Yuan (2017)* found significantly elevated concentrations of Mo in cores of Lake Erie after 1950 attributed to anthropogenic sources.

*Cadmium*

*Contaminant status in the Great Lakes:* Yes, listed as a sediment contaminant or water contaminant in 11 AOCs in the Great Lakes (*Hartig & Thomas, 1988*).

*Dominant sources:* Industrial releases.

*Background and relevance to the Great Lakes:* Cadmium (Cd) is a toxic element on the United Nations Environmental Program's list of top 10 hazardous pollutants and is a priority pollutant according to the United States Environmental Protection Agency

(*Emsley, 2002*; *Mudhoo, Garg & Wang, 2012*). It is the 65th most abundant element in the Earth's crust at 0.1 ppm. In soil, Cd averages 1 ppm, though polluted soil has been found to contain up to 1,500 ppm. Cd is thought to be completely non-essential for biological function (*Smolders & Mertens, 2013*; *Tchounwou et al., 2012*). Cd bioaccumulates and is a particular concern in plants, where it accumulates in leaves and is consumed by animals or humans (*Nagajyoti, Lee & Sreekanth, 2010*). It is also believed to be carcinogenic. Cd is used in electroplating steel and in rechargeable nickel-Cd batteries. It is a known atmospheric contaminant, being released by smelting, fossil fuel burning, and waste incinerations (*Norton et al., 2007*). Due to greater recognition of its toxicity and increased regulations, anthropogenic Cd emissions have decreased over 90% in the last 50 years since elemental alternatives were identified in the 1960s (*Agency for Toxic Substances & Disease Registry (ATSDR), 2012a*). Cd was found to be anthropogenically enriched in sediment cores throughout the Great Lakes (*Förstner, 1976*; *Goldberg et al., 1981*; *Kemp & Thomas, 1976a*, *1976b*; *Kemp et al., 1978*; *Marvin et al., 2003*; *Reavie et al., 2005*; *Walters, Wolery & Myser, 1974*; *Yuan, Depew & Soltis-Muth, 2014*; *Yuan, 2017*). *Förstner (1976)* attributed Cd in sediment cores from Lake Erie to industrial discharges and specifically found high enrichments in the late 1940s corresponding to the growth of the Cleveland electroplating industry. *Reavie et al. (2005)* associated Cd in a Lake George sediment core with production of alloys, metal coating processes, and leather tanning in the area of Sault Ste Marie. They found decreases in the sediments since the late 1950s, when Northwestern Leather Company closed.

*Hafnium*
*Contaminant status in the Great Lakes:* None known.
*Dominant sources:* Zr containing minerals and soil dust.
*Background and relevance to the Great Lakes:* Hafnium (Hf) is the 45th most abundant element in the Earth's crust at 5.3 ppm. In soils, it averages around 5 ppm, ranging from 2 to 20 ppm. Hafnium has no known biological role or toxicity. Hf is used as a control rod for nuclear reactors and submarines because it effectively absorbs neutrons, but otherwise it has limited applications. It is not known to be an environmental threat (*Emsley, 2002*; *Hampel, 1968b*). *Kerfoot & Robbins (1999)* found Hf concentrations to be moderately enriched in the Point Mills stamp sands on Lake Superior.

*Tantalum*
*Contaminant status in the Great Lakes:* None known.
*Dominant sources:* Occurs with Nb and the rare-earth elements in clay and shale.
*Background and relevance to the Great Lakes:* Tantalum (Ta) is the 51st most abundant element in the Earth's crust at an average of 2 ppm. It is insoluble and there is virtually no Ta in natural waters (*Emsley, 2002*), so in geochemical records it would be expected to be associated with mineral particulates. Ta has no biological role and is not known to have toxicity properties, hence it is widely used in surgical tools (*Wehrmann, 1968*). It is an essential metal in modern society and is also used for corrosion resistance,

microelectronics, and high-strength, low-alloy steel (*Mackay & Simandl, 2014*). This element has not been studied in the Great Lakes.

*Mercury*

*Contaminant status in the Great Lakes:* Yes, related to coal fired power plants, mining, and other industries.

*Dominant sources:* Atmospheric.

*Background and relevance to the Great Lakes:* Mercury (Hg) is relatively rare; at 50 ppb in the Earth's crust, it is the 68th most abundant element (*Emsley, 2002*). In soil it averages 0.01–0.5 ppb, though contaminated soils have up to 0.2 ppm. Despite its overall rarity, it is an element of major concern because its toxicity at very low concentrations threatens environmental and human health. Because harmful levels of Hg can be very low, analytical techniques for Hg require an especially low detection limit, much lower than that required for most other metallic elements. Hg has no known biological role but is present in all life. It is a potent neurotoxin that accumulates and magnifies in aquatic food webs (*Sherman et al., 2015*). Hg is a pollutant throughout the Great Lakes, and most water bodies in the Great Lakes region have fish consumption advisories due to Hg accumulation in tissues (*Risch et al., 2012*). As a contaminant of concern, much research has been performed on Hg levels in Great Lakes sediment and sediment cores (Table 1). Sediment cores reveal Hg increases beginning in the mid-1800s and reaching a maximum in the 1970s, decreasing in more recent times with increasing regulation and phasing out of most uses in modern industry (*Steinnes, 2013*). In modern times, Hg is released into the atmosphere primarily by coal-fired power plants, metal processing, and waste incineration (*Chambers et al., 2016*; *U.S. Department of the Interior (USDOI) & U.S. Geological Survey (USGS), 2017*). Historic Hg atmospheric emissions had peaks in 1879 and 1920 from gold and silver mining (*Pirrone et al., 1998*). Diffuse atmospheric sources are thought to dominate Great Lakes Hg, but there have been a number of localized point sources. Around the Keweenaw Peninsula (Superior), Hg loading was greatly enhanced during the height of Cu mining between 1880 and 1930 due to its presence in tailing piles, smelters, mills, and parent ores (*Kerfoot et al., 1999*, *2004*, *2016*). High loadings were found near pulp and paper plants near Thunder Bay, Nipigon, and Munising (*Fitchko & Hutchinson, 1975*; *Kemp et al., 1978*). In Lake Erie, major increases in Hg releases were seen around 1939 and 1954, corresponding with establishment and growth of two chlor-alkali facilities (*Azcue, Rosa & Mudroch, 1996*; *Wolery & Walters, 1974*).

*Lanthanides*

*Dominant Sources of Rare Earth Elements:* The term "rare earth" is somewhat of a misnomer, as they are not actually rare on Earth but they do appear together. The bulk of the lanthanides are found in eroded material in clay and shale, such as that in the Paleozoic province surrounding Lakes Michigan, Erie, and Ontario (Fig. 1). A very small percent is dissolved into solution. These elements take part in almost no chemical processes. Mobilization may occur during weathering but not likely during erosion and transport. Overall, there are minimal post-depositional changes in sedimentary records (*Fleet, 1984*).

*Lanthanum*

*Contaminant status in the Great Lakes:* None known but potential.

*Dominant sources:* Weathering of clay and shale.

*Background and relevance to the Great Lakes:* Lanthanum (La) is the 28th most abundant element at 32 ppm in the Earth's crust and approximately 26 ppm in soil (*Emsley, 2002*). La has been shown to be toxic in a number of organisms, and anthropogenic loadings have been found in several German rivers (*Kulaksız & Bau, 2011b*). In sediment cores of the Great Lakes, *Reavie et al. (2005)* and *Shimp et al. (1971)* found little or no variation in concentration from pre-industrial sediments to modern times. *Olivarez, Owen & Long (1989)* and *Neustadter, Fordyce & King (1976)* noted that the concentration of La in Great Lakes sediments was controlled by inputs of crustal sediments. However, *Herrmann et al. (2016)* highlight La's potential role as an emerging contaminant of concern with its use in innovative technologies such as super-conductors, catalysts, lasers, and batteries, and an increase in releases to aquatic environments.

*Cerium*

*Contaminant status in the Great Lakes:* None known but potential.

*Dominant sources:* Weathering of clay and shale.

*Background and relevance to the Great Lakes:* Cerium (Ce) is the 25th most abundant element in the Earth's crust at 68 ppm, averaging 50 ppm in soil (*Emsley, 2002*). Ce is the most abundant of the rare earth elements. It has no biological role and, like all rare earth elements, has low acute toxicity (*Gschneidner, 1968*). Ce is found in urban environments due to its use in welded metal plating (*Chambers et al., 2016*). It has found more modern uses in long-life, low-energy lightbulbs and as a vehicular exhaust cleaner. Ce has been suggested as a reference element in the sediment to account for soil influences so that the percentage of actual anthropogenic influx can be measured. Ce has a radioactive isotope, $^{144}$Ce, which is produced in nuclear reactors and was detected in Lakes Superior and Huron in 1974 and 1975 (*Durham & Joshi, 1984*; *Tracy & Prantl, 1983*). *Reavie et al. (2005)* found paleolimnological Ce concentrations in a Lake George core did not change over the most recent two centuries.

*Praseodymium*

*Contaminant status in the Great Lakes:* None known.

*Dominant sources:* Weathering of clay and shale.

*Background and relevance to the Great Lakes:* Praseodymium (Pr) is the 39th most abundant element in the Earth's crust at 9.5 ppm. It is a rare earth element and occurs in minerals exclusively with other members of the group. There is no documented environmental threat from this element. It can also exist as a fission product of U, Th and Pu (*Eyring, 1968*; *Emsley, 2002*).

*Neodymium*

*Contaminant status in the Great Lakes:* None known.

*Dominant sources:* Weathering of clay and shale.
*Background and relevance to the Great Lakes:* Neodymium (Nd) is the 26th most abundant element in the Earth's crust and the 2nd most abundant rare-earth element after Ce. Nd is almost as abundant as Cu at 38 ppm in the Earth's crust and around 20 ppm in soils. It has no known biological role or environmental threat (*Emsley, 2002*).

### Samarium

*Contaminant status in the Great Lakes:* None known but potentially an emerging contaminant of concern.

*Dominant sources:* Weathering of clay and shale.

*Background and relevance to the Great Lakes:* Samarium (Sm) is the 40th most abundant element in the Earth's crust at 8 ppm. Sm is found with other rare-earth elements (*Emsley, 2002*). Sm has no known biological role, though anthropogenic Sm was found to be bioavailable to freshwater clams (*Merschel & Bau, 2015*), and toxicity is uncertain. Sm is an emerging contaminant of concern in river systems. It is used for high-strength permanent magnets and for control rods in nuclear reactors. In the Rhyne River in Germany, a significant amount of Sm pollution was found downstream from a production plant for fluid catalytic cracking (*Kulaksız & Bau, 2013*). No pollution is currently known in the Great Lakes region, and in sediment cores from Lake George, *Reavie et al. (2005)* found no changes in Sm concentration over two centuries, indicating solely natural sources.

### Europium

*Contaminant status in the Great Lakes:* None known.

*Dominant sources:* Weathering of clay and shale.

*Background and relevance to the Great Lakes:* Europium (Eu) is the 50th most abundant element in the Earth's crust at 2 ppm. It is one of the most abundant of the rare earth elements. It appears to have no biological role or pose any environmental threat (*Emsley, 2002*). It is the most reactive of the rare earth metals and is used for the red phosphor in cathode ray color televisions and fluorescent lamps (*Thompson, 1968*). *Reavie et al. (2005)* found Eu in their sediment cores did not vary with sediment depth or composition and *Neustadter, Fordyce & King (1976)* considered atmospheric Eu over the Great Lakes to be soil derived.

### Gadolinium

*Contaminant status in the Great Lakes:* None known but potentially an emerging contaminant of concern.

*Dominant sources:* Weathering of clay and shale.

*Background and relevance to the Great Lakes:* Gadolinium (Gd) is the 41st most abundant element in the Earth's crust. It is one of the more abundant rare earth elements at an average of 8 ppm. No acute health effects are known, but potential long-term effects of low-level exposure are possible (*Kulaksız & Bau, 2011a*). Gd has been used as a contrasting agent in magnetic resonance imaging since 1988. It is not removed by wastewater treatment plants and is an emerging micropollutant of concern in surface waters in densely populated areas. Gd chelates can be used as tracers for other emerging microcontaminants that are not removed in wastewater treatment processing, such as steroids and
pharmaceuticals. *Bau, Knappe & Dulski (2006)* found increased Gd levels in the surface water of Lake Erie and the Niagara River, suggesting input from the urban area of Buffalo.

*Terbium*
*Contaminant status in the Great Lakes:* None known.
*Dominant sources:* Weathering of clay and shale.
*Background and relevance to the Great Lakes:* Terbium (Tb) is the 57th most abundant element in the Earth's crust at 1 ppm. It is one of the rarer rare-earth elements and occurs in minerals with the others of that group. Tb does not have many uses, as it is very expensive to acquire and separate from the other rare-earths, but it is used in lasers and low-energy lightbulbs (*Emsley, 2002*). It has no known biological role and is not considered an environmental threat.

*Dysprosium*
*Contaminant status in the Great Lakes:* None known.
*Dominant sources:* Weathering of clay and shale.
*Background and relevance to the Great Lakes:* Dysprosium (Dy) is the 42nd most abundant element in the Earth's crust at 6 ppm. It is a rare-earth element and is found associated with the other rare-earth elements in minerals. It has no biological role and low acute toxicity (*Emsley, 2002*; *Powell, 1968*). *Kerfoot & Robbins (1999)* found that Dy showed modest enrichment in the stamp sands submerged in waterways of the Keweenaw Peninsula over background concentration. Due to its extremely low background level in Lake Michigan, *McCown (1979)* used Dy to tag experimental simulated oily waste in order to determine its motion in freshwater.

*Holmium*
*Contaminant status in the Great Lakes:* None known.
*Dominant sources:* Weathering of clay and shale.
*Background and relevance to the Great Lakes:* Holmium (Ho) is one of the rarer rare-earth elements at 1.4 ppm in the Earth's crust and ~1 ppm in soils. It has no biological role and poses no environmental threat. Ho has the highest magnetic strength of any element (*Emsley, 2002*). In a sediment core of Lake Superior, *Korda et al. (1977)* found Ho concentrations varied between 0.7 and 1.4 ppm and had no apparent anthropogenic component.

*Erbium*
*Contaminant status in the Great Lakes:* None known.
*Dominant sources:* Weathering of clay and shale.
*Background and relevance to the Great Lakes:* Erbium (Er) is one of the more abundant rare-earth elements at 4 ppm in the Earth's crust and ~1.6 ppm in soil (*Emsley, 2002*). It has no biological role and poses no environmental threat. *Korda et al. (1977)* found Er concentrations between 1.8 and 3.1 ppm in their core of Lake Superior.

*Thulium*
*Contaminant status in the Great Lakes:* None known.

*Dominant sources:* Weathering of clay and shale.

*Background and relevance to the Great Lakes:* Thulium (Tm) is the second rarest rare-earth element after promethium at 0.5 ppm in the Earth's crust and an average of 0.5 ppm in soil. *Mayfield & Fairbrother (2015)* examined the concentrations of rare-earth elements in the tissues of various freshwater fish, and Tm was the only element that was not detectable in the species examined.

*Ytterbium*

*Contaminant status in the Great Lakes:* None known.

*Dominant sources:* Weathering of clay and shale.

*Background and relevance to the Great Lakes:* Ytterbium (Yb) is one of the rarer rare-earth elements at 3 ppm in the Earth's crust and around 2 ppm in soil. It has no biological role and does not pose an environmental threat (*Emsley, 2002*). In an undated sediment core from western Lake Superior, *Korda et al. (1977)* found concentrations of Yb ranged from 1.6 to 3.1 ppm. In cores of Lake George, *Reavie et al. (2005)* found Yb concentrations did not vary over time or by sediment composition, indicating that natural sources were dominant, although *Kerfoot & Robbins (1999)* found moderate enrichment of Yb in the stamp sands of Lake Superior.

*Lutetium*

*Contaminant status in the Great Lakes:* None known.

*Dominant sources:* Weathering of clay and shale.

*Background and relevance to the Great Lakes:* Lutetium (Lu) is one of the rarer of the rare earth elements at 0.5 ppm in the Earth's crust. Lu has no known biological role and is thought to pose no environmental threat. Lu is found in minerals that contain all of the lanthanides (*Emsley, 2002*). In their sediment cores of Lake George, *Reavie et al. (2005)* found that Lu did not vary with sediment depth or composition, showing no anthropogenic changes in Lu downstream from Lake Superior.

### Basic metals

*Aluminum*

*Contaminant status in the Great Lakes*: Some historical contamination associated with taconite tailings in the western part of Lake Superior.

*Dominant sources*: Feldspars and clay minerals, runoff and erosion.

*Background and relevance to the Great Lakes*: Aluminum (Al) is the most abundant metal (7%) in the Earth's crust (*Khan et al., 2013*). Al has a strong sorption tendency, binding strongly to clay minerals and natural organic matter (*Lum & Leslie, 1983*; *Driscoll, 1985*). Al is not known to be a nutrient for any plants or animals (*Driscoll & Schecher, 1990*). Elevated concentrations of ionic Al are also important because Al acts as a buffer and influences the cycling of elements such as P and C through adsorption and coagulation reactions (*Driscoll & Schecher, 1990*). It is highly insoluble and unreactive under most conditions, but below a pH of 6 Al can dissolve from substrates and become available in its ionic form, at which point it becomes bioavailable and can be toxic (*Bache, 1986*). Ionic Al

is released in acidic waters often related to acid rain and acid mine drainage (*Agency for Toxic Substances & Disease Registry (ATSDR), 2008*). Fluxes in total Al in lakes and streams are virtually all associated with sediment transport and runoff and erosion (*Driscoll, 1989*; *Norton et al., 1992*; *Long et al., 2010*). Lakes are sinks for Al because of their acid neutralizing capacity (*Driscoll, 1989*), so toxic levels of Al in the Great Lakes are unlikely. Spikes in overall Al levels in sediment cores are often associated with land clearing and logging (*Selig, Leipe & Dörfler, 2007*), which expose clay/Al-bearing soils to erosion. In Lake Superior, both *Kemp et al. (1978)* and *Shaw Chraïbi et al. (2014)* saw an increase in Al correlated with the input of taconite tailings between 1955 and 1967. According to *Oxberry, Doudoroff & Anderson (1978)*, Al from fresh taconite tailings is not soluble in Lake Superior waters, which are slightly alkaline and have a pH generally above 7.3, so there was not likely to be toxicity effects. In sediment cores taken from Lakes Ontario, Erie and Huron, *Kemp & Thomas (1976b)* found the concentration of Al to remain generally uniform over the period from pre-settlement to the early 1970s.

*Gallium*
*Contaminant status in the Great Lakes:* None known but possible.
*Dominant sources:* Clay minerals, coal fly ash.
*Background and relevance to the Great Lakes:* Gallium (Ga) is more abundant than Pb in the Earth's crust, averaging 18 ppm, but unlike Pb, Ga is not concentrated in any minerals, so it is more difficult to obtain using industrial processes (*Emsley, 2002*). Some Ga is incorporated in clay minerals, and it can substitute for Al in sediments (*Nussmann, 1965*). Ga is recovered as a byproduct of processing Zn ores and bauxite and is used in microchips, lasers, solar cells, and LEDs (*U.S. Department of the Interior (USDOI) & U.S. Geological Survey (USGS), 2017*). Ga has no biological role, and *Reid, Rengel & Smith (1996)* demonstrated Ga toxicity to green algae, though they found it to be less toxic than Sc and Al. Ga averages 7 ppm in coal, but in coal ash, Ga averages 100 ppm (*Blowes et al., 2014*). In surficial sediments from Lake Michigan, *Cahill (1981)* found Ga concentrations from 0 to 20 ppm. *Nussmann (1965)* found Ga in Lake Michigan averaged 19–22 ppm, and Ga content of Lake Superior sediments averaged 13.2 ± 5.4 ppm, seeming to increase slightly near the top of the sediment cores, though this increase was not statistically significant.

*Tin*
*Contaminant status in the Great Lakes:* Yes, attributed to industry and fossil fuel emissions.
*Dominant sources:* Largely atmospheric emissions from smelting.
*Background and relevance to the Great Lakes:* Tin (Sn) is the 49th most abundant element in the Earth's crust at 2 ppm. It averages around 1 ppm in soil. Sn is insoluble and its ore is very resistant to weathering, so the amount of Sn in soil and natural waters is low (*Emsley, 2002*). There is some evidence that it might be an essential micronutrient in mammals, though it is also a part of compounds which are enzyme disruptors in aquatic organisms (*Alloway, 2013b*). Some of its uses include plating cans and containers, solder, and fungicides (*Blowes et al., 2014*), and it has become an anthropogenic contaminant in

natural systems. *Robbins (1980)* found Sn to be 2nd only to Hg in terms of enrichment in Southern Lake Huron. *Biegalski & Hopke (2004)* determined that Sn in the atmosphere in the Great Lakes region was attributable to smelting. *Goldberg et al. (1981)* traced Sn sources in Lake Michigan to coal, coke and fuel oil, noting that it had similar profiles in sediment cores to charcoal. *Yuan (2017)* categorized Sn with the most enriched elements in Lake Superior sediments.

*Lead*

*Contaminant status in the Great Lakes:* Yes, related to fossil fuel emissions, mining.
*Dominant sources:* Leaded gasoline, shoreline bluffs of Lake Erie.
*Background and relevance to the Great Lakes:* Lead (Pb) is not a particularly abundant constituent in the Earth's crust; at 14 ppm, it is the 36th most abundant element (*Emsley, 2002*). Pb is used as a proxy for human activity because it has a relatively low background concentration in sediments, averaging 23 ppm worldwide, with 18 ppm in Lake Erie (*Azcue, Rosa & Mudroch, 1996*; *Pompeani et al., 2013*). Pb is a contaminant of particular interest because it is a cumulative poison that can damage organisms at low concentrations (*Forsythe & Marvin, 2009*). Releases from anthropogenic sources, most notably as a fuel additive, dwarf those from natural sources (*Agency for Toxic Substances & Disease Registry (ATSDR), 2007c*). The main current anthropogenic sources of Pb in the environment are mining and smelting, manufacture of Pb containing products, combustion of coal and oil, and waste incineration. As for all metals, natural sources of Pb from catchment soil and bedrock include fluvial and shoreline bluff erosion, such as that detailed by *Kemp et al. (1976)* and *Ritson et al. (1994)* in Lake Erie. A great number of sediment cores have examined Pb concentrations in the Great Lakes over time (Table 1). It is enriched in surface sediments of all of the lakes due to anthropogenic activity. *Rossmann, Pfeiffer & Filkins (2014)* found the following trends in Lake Michigan: in 1850 Pb levels increased above background levels correlated with coal and gas consumption. Pb loading increased with industrial activity during World War II and began to decline with the Clean Air Act of 1970 and the phasing out of leaded gasoline. According to *Rossmann, Pfeiffer & Filkins (2014)*, Lake Michigan loading in 1995 was 52% lower than peak loading in 1970. However, in the Sandusky Bay of Lake Erie, *Yuan (2017)* found only a 36% reduction from 1970 to 2010. *Yohn et al. (2004)* found that anthropogenic loading from the 1930s to the 1970s was mainly driven by regional sources such as leaded gasoline and since the 1970s has been much more closely correlated to local population density. This was attributed to legacy pollution and Pb cycling, since Pb does not degrade in the sediments. *Pompeani et al. (2013, 2015)* found periods of increased Pb loading in the Keweenaw Peninsula 8000–5000 years ago which they associated with Cu mining.

## Semimetals

*Silicon*

*Contaminant status in the Great Lakes:* Possibly some as dust.
*Dominant sources:* Rocks, minerals, soils, dust.

*Background and relevance to the Great Lakes:* Silicon (Si) is the second most common element in the Earth's crust after oxygen. It makes up 28% of Earth's crust and is the key component of soil, occurring in almost all minerals (*Emsley, 2002*). Si is the third most abundant element in the human body and is involved in bone mineralization. It is also a carcinogen and is responsible for silicosis, a disease that occurs from inhalation of Si dust (*Martin, 2013*). Si never occurs in sediments uncombined; it occurs either with oxygen as $SiO_2$ (silica) or with oxygen and metals as silicates (*Lanning, 1968*). Si is relatively unreactive and insoluble. Increases in Si concentration come from more intensive weathering of silicates and are associated with warmer or wetter conditions. According to *Chambers et al. (2016)*, Si levels are relatively high in some urban waters due to its use in building materials such as clay and sandstone. Si was found to be a conservative element that remained relatively constant in sedimentary records from Lakes Ontario, Erie, Huron, and Superior (*Kemp et al., 1976*, *1978*; *Kemp & Thomas, 1976a*, *1976b*). *Mudroch (1985)* considered Si to be of terrigenous origin in the Detroit River, and *Holsen et al. (1993)* also considered it a crustal element. In their core of the western basin of Lake Superior, *Shaw Chraïbi et al. (2014)* found peaks in Si around 1960 related to the period of taconite processing waste discharge in Silver Bay. Minerals containing silica comprise more than 95% of the Earth's crust (*Agency for Toxic Substances & Disease Registry (ATSDR), 2017b*). Silica, in the form of quartz, is extremely resistant to weathering. In the form of other minerals, $SiO_2$ is able to dissolve in water to form bioavailable silicic acid, an important compound for biological use by some plants and algae. In the Great Lakes, there is an inverse relationship between $SiO_2$ concentration and spring diatom population size, as diatoms take up all available $SiO_2$ from the water column (e.g., Lake Erie; *Reavie et al., 2016*). As the main component in cell walls of diatoms, biogenic silica is a critical compound for algal communities. *Schelske et al. (1987)* used $SiO_2$ concentrations in Lakes Michigan, Erie, and Ontario as indices of eutrophication because as P loading increased, diatom production increased, eventually leading to $SiO_2$ depletion and limited diatom production. In modern times, the inverse is occurring in Lake Ontario, where $SiO_2$ levels are increasing as a symptom of the proliferation of invasive zebra and quagga mussels leading to decreasing diatom populations (*Dove, 2009*).

**Arsenic**

*Contaminant status in the Great Lakes:* Yes, listed as a sediment contaminant or water contaminant in 10 AOCs in the Great Lakes (*Hartig & Thomas, 1988*).

*Dominant sources:* Mining and other industry, fossil fuel burning.

*Background and relevance to the Great Lakes:* Arsenic (As) averages around 1.5 ppm in the Earth's crust and 1–10 ppm in soil (*Emsley, 2002*). It is a deadly poison and is carcinogenic (*Plant et al., 2014*). As occurs naturally in soil and airborne dust and is associated with fine particles (*Sweet, Weiss & Vermette, 1998*). Anthropogenic releases of As are much higher than natural releases and As ranks high on hazardous substances lists (*Agency for Toxic Substances & Disease Registry (ATSDR), 2007a*). Arsenic's primary use is as a wood preservative to prevent rot, though it was previously used as a component of pesticides and fungicides. It is also associated with mining and smelting of Cu, Pb, Zn and Sn and is

present in coal and phosphate fertilizers. In the Great Lakes, As is associated with ferromanganese nodules in Lake Michigan, particularly near Green Bay, and is released with weathering and erosion. Increased As concentrations in aerosols over Lake Huron and Lake Michigan have been related to fossil fuel use and smelters (*Biegalski & Hopke, 2004*; *Shahin et al., 2000*). *Nriagu (1983)* measured changes in As concentrations near smelters at Sudbury Ontario that mirror Cu and Ni production. In Lake Superior, As was found increased five fold in taconite, mine tailings, and the Freda stamp sands compared to a control area (*Korda et al., 1977*). *Reavie et al. (2005)* found anthropogenically enriched As concentrations in post-settlement sediments of Lake George, which they attributed to herbicide use. *Förstner (1976)* and *Walters, Wolery & Myser (1974)* saw major increases in Lake Erie sediments between 1939 and 1955, which they related to the growth of industry related to WWII and the Korean conflict.

*Antimony*

*Contaminant status in the Great Lakes:* Yes, related to mining and other industry.

*Dominant sources:* Anthropogenic sources.

*Background and relevance to the Great Lakes:* Antimony (Sb) is the 63rd most abundant element in the Earth's crust at 0.2 ppm. It has an average abundance of approximately 1 ppm in soil (*Emsley, 2002*). Sb has relatively low mobility in soils, has no biological role, and is highly toxic. While naturally rare, Sb is anthropogenically enriched in soils and lakes. According to *Clemente (2013)*, Sb is enriched on the order of 70 times compared to background concentrations in the United States. Sb contamination comes from mining and other activities that result in releases of metal particles such as shooting ranges and brake dust (*Agency for Toxic Substances & Disease Registry (ATSDR), 2017a*). It is a priority pollutant according to the USEPA (*Chen et al., 2003*; *Filella, Belzile & Chen, 2002*). *Reavie et al. (2005)* found Sb enriched in the sediment of Lake George related to the production of alloys and metal coating processes in the adjacent watershed. *Biegalski & Hopke (2004)* found Sb highly enriched from crustal sources in aerosols over the north shore of Lake Huron. *Robbins (1980)* determined Sb was the fourth most enriched pollutant in the surface sediments of Lake Superior, after Hg, Sn and Pb. In Lake Erie, initial major increases in Sb concentrations occurred between 1948 and 1953 due to industrial uses and more diffuse sources (*Walters, Wolery & Myser, 1974*).

### Nonmetals

*Phosphorus*

*Contaminant status in the Great Lakes:* Yes, as a nutrient pollutant.

*Dominant sources:* Agriculture, industry, sewage.

*Background and relevance to the Great Lakes:* Phosphorus (P) is one of the most widely distributed elements. At 1,000 ppm in the Earth's crust, P is the 11th most abundant element and averages 1.5 ppb in surface waters. P is necessary for all life, though in its elemental form P is highly poisonous and reactive. In nature, P exists primarily as $PO_4$ (*Emsley, 2002*; *Whaley & Currier, 1968*). As a limiting nutrient for algal growth, anthropogenic P loading is a problem in natural waters such as the Great Lakes because too

much P can support cultural eutrophication (*Manning, Lum & Birchall, 1983*). Sources of anthropogenic P in urban environments include laundry detergent, fertilizer, food waste and sewage (*Chambers et al., 2016*). Primarily as a response to increasing concerns about eutrophication, the United States and Canada implemented The Great Lakes Water Quality Agreement (*Canada & United States of America, 1972*). Both countries implemented P removal programs aimed at industrial and municipal point sources and reductions in phosphates in detergents (*Baker et al., 2014*). P levels have been a particular concern in Lakes Erie and Ontario. Lake Superior, the most oligotrophic of the Great Lakes, is thought to be P limited (*Baehr & McManus, 2003*). Analyzing sedimentary P can be a poor indicator of actual P levels and eutrophication due to variable retention and post-depositional diagenesis. For this reason, diatom-inferred P levels are used to give a better picture of eutrophication, as they provide a surrogate measure of P levels that are not subject to these factors (*Reavie, Heathcote & Chraïbi, 2014*). In an Eastern core of Lake Superior, *Shaw Chraïbi et al. (2014)* found P levels peaked between 1990 and 1994. In the Western Core they found a peak in P around 1960, which they associated with taconite processing near Silver Bay. In sediment cores from Lakes Ontario, Huron and Erie, *Kemp & Thomas (1976a*, *1976b*) found increases in P related to anthropogenic loading.

### Actinide series

#### Thorium

*Contaminant status in the Great Lakes:* Yes, part of the Port Hope Area of Concern.
*Dominant sources:* Naturally ubiquitous and a byproduct of U mining and processing.
*Background and relevance to the Great Lakes:* Thorium (Th) is a typically radioactive metal which is ubiquitous in the environment, occurring naturally in the Earth's crust at concentrations between 8 and 12 ppm (*Agency for Toxic Substances & Disease Registry (ATSDR), 1990*). Th is strongly adsorbed by clays, with a mean concentration of 30 ppm in clay minerals (*Langmuir & Herman, 1980*). Th is also generated as a byproduct of U mining and processing. This can result in anomalously large, localized concentrations of Th in the environment that may be harmful. In the Port Hope Area of Concern (north shore of Lake Ontario), water quality and environmental health were threatened by Th contamination, a result of poor operation and waste management practices of Eldorado Mining and Refining between 1933 and 1953 (*Government of Canada, 2017*). An estimated 85,000–95,000 cubic meters of sediment containing radioactive material is present in this harbor in Ontario on the north shore of Lake Ontario. *Sakaguchi et al. (2006)* examined Th in the sediment records of Lake Baikal. They used Th as a proxy for terrestrial materials, as it has low solubility and is resistant to weathering. They found that it was enriched in the fine grain components of their core. In cores of Lake George, *Reavie et al. (2005)* found Th concentrations largely invariant with sediment composition and depth.

#### Uranium

*Contaminant status in the Great Lakes:* Yes, part of the Port Hope Area of Concern.
*Dominant sources:* Ubiquitous, also associated with mining and nuclear power and weapons generation.

*Background and relevance to the Great Lakes:* Uranium (U) is present at 2 ppm in the Earth's crust and 0.7–11 ppm in soil. In agricultural soil with phosphate fertilizer, the soil concentration averages 15 ppm. Coal combustion is also a significant source of U in the environment. U is ubiquitous and is found in most rocks and soils, though it is higher in rocks that are enriched in Na, K, or Si (*Seaborg, 1968*). It is naturally dominated by isotopic forms; $^{238}$U accounts for 99% of natural U by mass. There is no known biological need for U, and it is both chemotoxic and a radiotoxic (*Alloway, 2013c*). U is mined and used for power generation and nuclear weapons, though it is naturally present in low concentrations in the Great Lakes as a result of weathering (*Ahier & Tracy, 1995*). Nearly all of the steps of nuclear power generation have occurred in the Great Lakes basin in the forms of mining, fuel preparation, power production, and waste management (*Ahier & Tracy, 1995*). In the Port Hope Area of Concern (north shore of Lake Ontario), water quality and environmental health were threatened by U contamination, a result of poor operation and waste management practices of Eldorado Mining and Refining between 1933 and 1953 (*Government of Canada, 2017*). An estimated 85,000–95,000 cubic meters of sediment containing radioactive material is present in this harbor in Ontario on the north shore of Lake Ontario. Contaminated soils from Port Hope have U concentrations of up to 258 ppm. Stable U concentrations have not been analyzed in sediment cores of the Great Lakes.

### A note on oxides

Several metals in aquatic systems are commonly present as detrital oxides. Fe and Mn oxides are particularly major sources of metals from river systems. Heavy metals are frequently sorbed to the surface of oxide molecules. Oxides may represent portions of an element that are not part of other rocks, that is, the actual mineral formations or ores. Some notable oxides include hematite ($Fe_2O_3$), an economically important iron ore that is found around the southern and northwestern shores of Lake Superior in Michigan, Wisconsin, and Minnesota, as well as ilmenite ($Fe_2TiO_3$), the major source of Ti (*Klein & Dutrow, 2008*).

## SUMMARY/DISCUSSION

Many studies have been devoted to analyzing inorganic geochemistry in the Great Lakes, and we have summarized metals as they relate to geochemistry and environmental stress in the lakes. Through these studies, natural conditions and anthropogenic impacts have been measured over time through the sourcing of metals from catchments, industry, agriculture, mining, and other human activities. Several studies also show reductions in these metals in the atmosphere, water, and lake sediment after legislation and mitigation efforts. With a few exceptions, it is difficult to pinpoint sources of contaminant metals due to their ubiquity in minerals comprising the bedrock and soils in the Great Lakes catchment. This comprehensive summary of Great Lakes-relevant metals serves as a key reference for the companion manuscript that details the implications of this review. This summary was a natural first step in developing background information for a

partner study on applied metals geochemistry throughout the Great Lakes (*Aliff et al., 2020*). The companion paper features an analysis of 11 sediment cores collected from all five Great Lakes. Taken together we believe that these two papers represent a thorough summary of the knowledge to date of geochemistry in sediment cores in the Great Lakes system.

## ACKNOWLEDGEMENTS

This document has not been subjected to the EPA's required peer and policy review and therefore does not necessarily reflect the view of the Agency, and no official endorsement should be inferred. Todd Anderson, John James Fielding, and two anonymous reviewers provided valuable comments on this manuscript.

### Funding

This research was supported by: (A) a grant to Euan Reavie from the U.S. Environmental Protection Agency under Cooperative Agreement GL-00E23101-2; and (B) State Special funding from Minnesota to the Natural Resources Research Institute. There was no additional external funding received for this study. The funders had no role in study design, data collection and analysis, decision to publish, or preparation of the manuscript.

### Grant Disclosures

The following grant information was disclosed by the authors:
U.S. Environmental Protection Agency under Cooperative Agreement: GL-00E23101-2.
Minnesota to the Natural Resources Research Institute.

### Competing Interests

The authors declare that they have no competing interests.

### Author Contributions

- Malachi N. Granmo conceived and designed the experiments, performed the experiments, analyzed the data, prepared figures and/or tables, authored or reviewed drafts of the paper, and approved the final draft.
- Euan D. Reavie conceived and designed the experiments, performed the experiments, analyzed the data, authored or reviewed drafts of the paper, and approved the final draft.
- Sara P. Post analyzed the data, prepared figures and/or tables, authored or reviewed drafts of the paper, and approved the final draft.
- Lawrence M. Zanko analyzed the data, prepared figures and/or tables, authored or reviewed drafts of the paper, and approved the final draft.

### Data Availability

    No raw data or codes were used in this literature review.

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
