# Peer review of "Metallic elements and oxides and their relevance to Laurentian Great Lakes geochemistry"

_PeerJ, doi:10.7717/peerj.9053_

## Round 0.1 · original submission · Major Revisions

There are some good reviewers comments for you to consider in revising your manuscript. Please address them and/or revise your literature review accordingly if appropriate.

Reviewer 1 ·

Basic reporting

A comprehensive review of trace metals in the Great Lakes, but useful as a general overview of trace metals as well. I suspect this will be valuable to a number of other researchers and recommend publication.

Experimental design

All relevant studies have been cited and summarized (to my knowledge).

Validity of the findings

No comment

Additional comments

I have mostly minor comments to do with word choice and the like. Some of the language is a little informal (e.g., line 100, 357, 590) so I suggest a careful review of that by the authors.
Line 31- Suggest rewording to “Metals are of particular interest because they record a history of pollution…”
Lines 40-41- Suggest briefly discussing why lakes that are not turbid are not ideal.
Line 132- “What this study shows…” Which study? The study being summarized in the review here? Or does this refer to the study Ouyang & Bartholic (2003)?
Lines 144-163- Organizationally, it makes more sense to me to put the natural sources and indicators of Li at the beginning of the section, followed by the anthropogenic ones.
Lines 236-238- Is it worth mentioning the impact of other nuclear events (e.g., Chernobyl, Fukishima)?
Line 292- Suggest rewording this sentence as initially I read it as stable concentrations of Ca up to 1955 (ppm), not 1955 AD.
Lines 340-342- These two sentences indicate very different spatial patterns of Ba pollution- worth explicitly pointing out I think.
Line 375- Suggest changing this unit from μg/m3 to ppb to be consistent with other units mentioned in the text.
Line 390- What is “plc”?
Line 558- Change “high densities” to “high concentrations”
Lines 558-559- Run-on sentence
Line 604- “Yes, but unclear”- this requires some further explanation but I don’t see it expanded upon in the following paragraph.
Lines 669-670- Disagreement in verb tenses.
Line 851- Worth briefly mentioning how Al influences the cycling of P and C
Lines 908-909- Suggest rewording as the structure of the sentence is awkward.
Line 936- Clarify that this is more intense weathering of silicates.
Table 2- Why does the average for sedimentary iron ores have a question mark after it? Also how are these “range usually reported” values calculated? Is this something the authors have personally observed based on some number of references (which need to be cited) or is this range based on someone else’s work (which needs to be cited)? I see that subscript “a” is a reference to Onishi (1969) and Boyle and Jonasson (1973) but I can’t find subscript “a” in the actual table.

Reviewer 2 ·

Basic reporting

no comment

Experimental design

The major strength of the study is the exhaustive literature review.

Validity of the findings

no comment

Additional comments

The manuscript by Aliff et al. entitled “Metallic elements and oxides and their relevance to Laurentian Great Lakes geochemistry” reviews available published literature regarding metal geochemistry in the Laurentian lakes. The outstanding strength of this paper is the exhaustive literature review, which is a valuable contribution. I do have some suggestions. Broadly speaking, the introduction needs improvement in the flow of the writing (see specific comments). Given the extremely long review of each element, the conclusion section seems very brief. I would suggest adding more to this section, including more implications, findings, or future directions of research. However, I do realize this is one paper of a two-part study. Perhaps these are found in the companion paper, but this needs to be specifically mentioned in this text (what exactly is in the contents of the companion manuscript and how does it relate to this study). Additionally, some discussions found in the geochemistry review do not seem to be related to lake geochemistry at all. For example, “[Mo] is generally toxicologically harmless in humans and other animals except ruminants, in which it causes a Cu deficit, which is reversible by taking Cu supplements (line 610).” It is unclear how this is relevant to lake geochemistry. I suggest either clarifying the relationship to lakes or removing all discussion unrelated to lake geochemistry here and other places in the text. More specific comments are below:

Line 21: “We expect that this review will serve as a key reference for…”. This wording is awkward. How can a future outcome be anticipated in a scientific paper? I would suggest deleting this sentence and moving down the sentence (line 14) to replace it, “We summarized available information on metals and their roles as geochemical indicators in the Great Lakes.”

Line 25: Remove “…on…” Change to “…because it can be used to estimate natural baseline, remediation targets…”

Line 26: I think to you need a comma after “epoch.”

Line 29: The sentence “Sediments can be collected as chronological profiles and by analyzing these sediments geochemical histories can be revealed” is way too complicated. How about the “Sediment cores can be collected and dated to reveal local geochemical histories.”

Line 31: “…hundreds of years…” Based on citations in this manuscript, this is most likely thousands of years.

Line 64: “…with both an eye to the complex… etc.” This is awkward wording for a scientific manuscript. Why not delete everything after “…extensively studied…”?

Lines 81-86: Why the quotes for such small portions of another text? You are citing the relevant work, right? Then they are not needed. Plus, quoting two sentences in one paragraph should not happen.

Line 113: Perhaps I am missing something, but it is not clear why only arsenic concentrations are shown in Table 2.

Line 137: Sometimes? When did you not use your best judgement? Did you always use your best judgement? If so, then say it (this is a long way of saying delete “sometimes”).

Line 197: I don’t believe Pompeani 2013 discussed K leaching into rivers from fertilizer applications. I would add an additional citation that specifically mentions the fertilizer mechanism for K.

Line 922: Broadly speaking, natural Pb in all lakes is derived from weathering geologic materials in the watershed (bedrock, soils, tills, etc.), along with relatively small amounts delivered in air (loess, wildfire, etc.). There is no need to specify fluvial or shoreline bluff erosion for Lake Erie. That is obvious. This is like saying “Lake Erie contains natural Pb because it is underlain with bedrock that weathers through time.”

Line 1049: Two uses of fertilizer in a row.

Line 1084: The end of the conclusions has to be re-written to not sound like the co-authors are anticipating an outcome from their research. For example, “But, as the first comprehensive summary of Great Lakes-relevant metals, we hope that this review will be helpful as a reference for future paleolimnological research. This summary was a natural first step in developing background information for a partner study on applied metals geochemistry throughout the Great Lakes.”

Why not change to something like this?

“This comprehensive summary of Great Lakes-relevant metals, serves as a key reference for the companion manuscript that details the implications of this review.”

·

Basic reporting

See general comments

Experimental design

See general comments

Validity of the findings

See general comments

Additional comments

Dear Editor and Authors,

Thank you for you submission. Clearly a lot of work has gone into collating a very large amount of information from very many texts. On this occasion I have decided to reject this article for publication, but I would strongly encourage the authors to resubmit the article once they have undertaken some changes.

Instead of going into fine detail I have included herein some broader changes, comments and suggestions which I believe would improve this manuscript to the point where it can be resubmitted for peer review. I hope what I say here is helpful and if the authors have any questions regarding the manuscript I am more than happy to answer them directly.

Generally the English is of a good standard, but does contain some errors, and I would encourage the authors to review this carefully before resubmission. Specific examples include: Line 117 ‘Compare that to the 1 to 2,900 ppm range shown in the table.’ this sentence doesn’t stand alone and examples of similar sentences appear throughout the text. In this case I would merge it with the sentence before in order to make it flow better; Lines 120 – 122 is another example of where I feel merging sentences would be beneficial; Line 194 ‘earth’s’ should have a capitol E and this mistake is repeated throughout the manuscript.

A more visual approach
It is always a problem to present large volumes of data in a concise way, this said I feel the authors approach can be made more concise by taking information out of the text and presenting in tabular form. This is especially true to information that is repeated for each individual element, for example background information about the element (abundance in the Earth’s crust), general environmental effects and or common health effects. Equally, including information such as the amount of sites and or studies a contaminant appears in and the age range, or maximum concertation would help to give the reader an immediate sense of the contamination scale.

In the same vein, presenting the data in more figures would help the reader and show case the author’s hard work. Other solutions may well be better, but, I could suggest perhaps presenting elements of concern (i.e. As, Hg, Pb) on a map per site and colour coding for abundance. The authors could also brake this down further by showing changes through time, i.e. showing abundances on a map for set times or periods, for example 1700, 1750, 1800 and so on. An example of this sort of figure but for pollen is fig. 2 in Williams et al, (2004) Late‐Quaternary vegetation dynamics in North America: scaling from taxa to biomes. Ecological Monographs, 74(2), 309-334, fig. 6 in Woodbridge et al. "Pan‐Mediterranean Holocene vegetation and land‐cover dynamics from synthesized pollen data." Journal of Biogeography 45, no. 9 (2018): 2159-2174. Fig. 2 in Moorhouse, et al. "Regional versus local drivers of water quality in the Windermere catchment, Lake District, United Kingdom: The dominant influence of wastewater pollution over the past 200 years." Global change biology 24, no. 9 (2018): 4009-4022 shows this for pollution in lakes across a catchment.

I would also like to see major industrial sources and other point sources of pollution on map, this is especially important for people who are not familiar with the great lakes area.

Added extra
I appreciate that it is beyond the scope of your original paper and that some of this may fall in the sister manuscript, however, I believe that by broadening the scope of the manuscript to include the following the authors will add more ‘value for money’ to the study. I also believe that it will help to concentrate the way in which the data is presented and help the manuscript to be more stand alone.

Statistical approach: Having done all the hard work to collect all this data I think it would a shame not to put it to work. I would like to see the authors statistically analyse elements distribution through time and space (different river catchments, lakes or geological areas), and if at all possible statistically evaluate a cause and effect relationship. To evaluate the first parameters I would suggest turning to the statistics of ecology and palaeopollen data which regularly evaluate the distribution of species in space and time. Test such as the 2 way ANOVA or general linear modelling can be good places to start. To analyse a cause and effect relationship the authors could also compare population increase for the area or agricultural produce with specific elemental abundance at different sites using a correspondence analysis for example PCA.

Comparison with environmental standards: There are various environmental sediment quality standards or threshold concentrations of specific pollutants above which public or ecological health is detrimentally effected. The authors already mention some of these in the text but I would like to see this highlighted further. This could for example be done in figure or tabular format, and show which sites exceed sediment quality standards and for which elements.

Thank you again for your submission and I look forward to seeing further from you.

Best Regards,
James Fielding, University of Southampton.

---

## Round 0.2 · accepted · Accept

Thank you for your efforts in revising your manuscript.

Reviewer 2 ·

Basic reporting

no comment

Experimental design

no comment

Validity of the findings

no comment

Additional comments

Nice work!